# The Impacts of Water Pollution Emissions on Public Health in 30 Provinces of China

**DOI:** 10.3390/healthcare8020119

**Published:** 2020-04-30

**Authors:** Zhen Shi, Shijiong Qin, Chenjun Zhang, Yung-ho Chiu, Lina Zhang

**Affiliations:** 1Department of International Trade and Economics, Business School, Hohai University, Changzhou 213022, China; 20051726@hhu.edu.cn; 2Department of Accounting, Business School, Hohai University, Changzhou 213022, China; 1863310220@hhu.edu.cn; 3Department of Economics, Soochow University, 56, Kueiyang St., Sec. 1, Taipei 10048, Taiwan; echiu@scu.edu.tw; 4Department of Business Administration, Business School, Hohai University, Changzhou 213022, China; 20191001@hhu.edu.cn

**Keywords:** wastewater treatment, health efficiency, two-stage dynamic SBM DEA, heavy metal concentration

## Abstract

China’s economy in recent decades has developed at a very rapid speed, as evidenced by its GDP jumping to second place in the world. Although utilization of domestic water resources has helped spur economic development, sewage discharge as an undesirable output has unfortunately caused many negative effects on human health, causing concern from all walks of life. Therefore, governments in China at all levels are committed to urban sewage treatment policies in order to reduce the negative impact of water pollution on society. While most existing studies have targeted the macro-level modes of economic development and environmental pollution, their selection of research objects is too narrow by failing to adequately consider China’s water pollution and the consequential national health crisis. This study takes cities in 30 provinces of China as the research objects and applies various influencing factors of urban wastewater treatment and health (as two stages) to the modified two-stage dynamic Slacks-Based Measures (SBM) Data Envelopment Analysis (DEA) model. The results reveal that the overall efficiency of each province is increasing and that the efficiency of the wastewater treatment stage is greater, thus contributing to overall efficiency. Conversely, the health stage’s efficiency is far lower than the wastewater treatment stage’s efficiency, which has a notably adverse effect on overall efficiency. In addition, most input-output variables need much improvement. Based on the findings herein, we offer specific suggestions to each province for improving sewage treatment capacity, the level of medical care, and the quality of national health.

## 1. Introduction

China’s economy has developed very rapidly to become the second largest one in the world ever since it opened itself up and initiated widespread reforms. However, at the same time, an increasingly prominent conflict has arisen between its economic development and ecological environment. Especially in the rapid development of urbanization in recent years, domestic water resources are becoming heavily polluted and a major health hazard. The problem of urban water pollution is now gravely restricting the sustainable development of the country’s economy.

With the dramatic development of the urban economy and the rising urban population year by year, water pollution caused by manufacturing and basic living needs has turned increasingly serious. Industrial areas are typically concentrated in the suburbs, and the large-scale machinery and equipment are discharging high amounts of sewage. Moreover, the sewage treatment efficiency of enterprises is low, which leads to secondary water pollution. The daily lifestyles of urban residents also produce large amounts of domestic wastewater, and hence, urban water pollution problems need to be urgently solved. Considering the above situations, the aim of this paper is to improve China’s urban sewage treatment capacity and the health of its residents.

In the existing research on environmental pollution, most scholars study the macro-level perspective of the relationship between the environment and the economy, but the economic impacts of environmentally friendly innovation and its knowledge externalities on productivity have attracted increasing attention from the research community. Aldieri et al. (2019) [1] presented empirical evidence of public policy strategies that support the dissemination of environmentally friendly technologies. The results of a systematic literature review showed that innovation activities on environmental issues can produce important knowledge spillovers. Aldieri et al. (2019) [2] discussed the relationship between enterprises’ knowledge resource strategy and green innovation. The results showed that the emphasis of environmental innovation has shifted from internal knowledge to external knowledge. Government policies that promote complementary and coordinated knowledge in the environmental field are able to contribute to greater knowledge transfer and more sustainable development. Studies have thus demonstrated the role of innovation in sustainable development from various perspectives.

Data envelopment analysis (DEA) is an important and widely used analytical method. Its basic idea is to determine the best practice boundary of effective decision-making units (DMUs) to cover all inefficient DMUs. The greatest advantage of using DEA is that there is no need to specify a production function, and that DEA can consider multiple inputs and outputs at the same time. Based on a modified two-stage dynamic Slacks-Based Measures (SBM) model, we study 30 provincial-level administrative units in China (not including Hong Kong, Macao, Taiwan, and Tibet autonomous region) and their overall efficiency, two-stage efficiency, and the efficiencies of the variables wastewater treatment and health (as two stages) from 2014 to 2017, employing scientific data that reflect their sewage treatment and health situation.

The contributions of this paper are mainly the following two aspects. First, we target the national level for the first time to study completed investments into wastewater treatment projects, sewage treatment plants, municipal sewage treatment capacities, and other indicators of specific dynamic efficiency in the 30 provinces. Accordingly, the paper provides reference data for the country and the provinces from the macro-level and microlevel aspects. Second, the research’s innovation is evaluating “wastewater treatment” and “health” in two stages. In the first stage, wastewater treatment efficiency, completed investments into wastewater treatment projects, and sewage treatment plants are the input variables, while municipal sewage treatment capacity is the desirable output variable, and total wastewater discharge, chemical oxygen demand (COD) concentration, and heavy metal pollutants’ equivalent concentration are undesirable output variables. On this basis, we can measure the efficiency of health in the second stage. In this stage, the number of health technicians and local fiscal medical and health expenditures are taken as input variables, while average life expectancy and carcinogenic risk are desirable and undesirable output variables, respectively. By comparing overall efficiency, two-stage efficiency, each component’s efficiency of 30 provinces in China, and combining them with China’s specific national conditions and regional economic differences such as human geography, we are able to observe the variables’ volatility, analyze the input-output efficiency values in greater detail, and put forward corresponding proposals to the provinces, which should provide a scientific basis for urban sewage treatment in the country.

## 2. Literature Review

According to previous references, the majority of scholarly research on urban sewage generated by firms’ production and humans’ lifestyles and their treatment can be carried out from the following three aspects.

### 2.1. The Impact of Water Pollution Caused by Urban Production and Living

Water pollution has negative impacts on the environment. Using a drink-y reservoir and an irrigation-t reservoir as research subjects, Deng et al. (2020) [3] found that metals can precipitate from water into sediment in 10–15 days, and both reservoirs are heavily contaminated with heavy metals (chromium, manganese, copper, zinc, Cd, mercury, and lead), which can be harmful to human health. The major anthropogenic sources of pollution are fuel mix and industrial mills (6.4%) and agricultural activities (38%) used for drinking; and fuel mix and industrial mills (4.9%), agricultural activities (32.9%), and mines and quarries used for irrigation (62.1%). Therefore, to reduce human health risks, freshwater should be stored 10–15 days before drinking or irrigation. Wojtkowska and Bojanowski (2018) [4] analyzed the impact of sewage and sewage management on the water quality of rivers by evaluating their eutrophication level. The research objects were the waterways of the Dluga, Pisia Gagolina, and Utrata rivers and the Srebrna stream. The results showed that the total phosphorus concentration in Utrata’s water is the lowest (mean 0.38 mg P/L), and the total phosphorus concentration in Diuga’s water is the highest (mean 2.8 mg P/L). The average concentration of orthophosphate is between 0.23 mg P/L (Dluga) and 0.45 mg P/L (Pisia Gqgolina). Moreover, the degree of phosphate pollution in the four river channels and the degree of eutrophication in their water are relatively high because the main sources of pollution in all rivers are wastewater from sewage treatment plants, leakage (damage or deliberate leakage) from septic tanks, surface runoff from agricultural areas and roads, and landfill leachate. According to the regional environmental protection watchdog, all watercourses are in poor ecological condition. The object of municipal sewage treatment has an important influence on water quality, and its pollutants can be carried from the sewage discharge place for a long distance.

Shi et al. (2019) [5] used a two-stage dynamic DEA model to study the impact of water pollution on the environment and national health. The authors divided variables into two stages. In the first stage (production), labor, energy, and water consumption were the input variables, and GDP was the desired output variable, while COD, CO_2_, and chromium emissions were undesired output variables. In the second stage (health), the local financial health expenditure, and the number of health technicians were input variables. The health index and the population mortality rate were the desired output and the undesired output variables, respectively. Fixed asset investment was selected as the carryover indicator in both stages. The findings showed that urban sewage damaged the sustainable development of the environment and economy to a certain extent and also dragged down the degree of national health.

Domestic and foreign scholars have conducted extensive research on the negative effects of urban sewage. Ho and Goethals’s (2019) [6] critical analysis of the contributions of individuals and subsets of sustainable development goals (SDGs) points to the global problem of lake and pond eutrophication caused by massive sewage discharge. Looking at sewage indicators, benthic cover measurements, macroalgae biometrics, and pollution scoring tools, Abaya et al. (2018) [7] studied Hawaiian coral reefs and detected that effluent from production may have contributed to the decline. Xin et al. (2019) [8] studied the impact of complex pollution sources on water quality in the Dengsha River basin of the city of Dalian, pointing out that the deterioration of water quality caused by excessive nutrient emissions from various point and non-point sources has been a global challenge. Li et al. (2019) [9] used the green bias technology progress model derived on the directional distance function to measure technology progress and its determinants obtained on inputs and outputs in 30 provinces and regions of China from 1999 to 2015. The results demonstrated that most of China’s provinces and regions overuse water in industrial production, and that output-oriented technological advances exacerbate the discharge of water pollutants which affect the green and sustainable development of the economy prior to the implementation of the 11th Five-year Plan (2006–2010).

Many scholars have taken sewage treatment plants (WWTPs) as the research object, finding that the carbon dioxide (CO_2_), methane (CH_4_), nitrous oxide (N_2_O), hydrogen sulfide (H_2_S), and other gases generated by WWTPs in the sewage treatment process also have many negative effects. Hu et al. (2019) [10] took 344 centralized sewage treatment plants out of 152 typical national industrial parks (NIPs) as the research target and established a set of calculation methods to measure the three main greenhouse gases of CO_2_, CH_4_, and N_2_O emitted by WWTPs in NIPs. Their main results are as follows: 5.64 million tons of CO_2_ equivalent (CO_2_-eq) were emitted, including 1.63 Mt CO_2_-eq on-site, 1.45 Mt CO_2_-eq off-site, and 2.56 Mt CO_2_-eq off-site related to sludge disposal. It can be seen that sewage treatment produced a large amount of greenhouse gases, causing a certain degree of impact on the environment and human health.

### 2.2. Municipal Sewage Treatment Methods

An et al. (2019) [11] discovered that national environmental laws and regulations to curb industrial wastewater came directly from the source and the structural impact because stringent environmental regulations can offset to some extent the inflow of the foreign direct energy-induced effect brought by the scale effect. In addition, the study also highlighted the increasing environmental investment and trade liberalization to improve the management of important industrial wastewater. Linge et al. (2012) [12] used datasets of 375 chemicals measured in reverse osmosis (RO) treated by WW (secondary wastewater), finding that dissolved organic carbon (DOC) in RO osmosis was between 3.7 and 10.7 mu g/L, attributable to at least one detected chemical, most of which was due to chemicals detected in less than 25% of samples. In conclusion, RO-treated WW is highly safe and can be regarded as an indirect source of drinking water.

Chen et al. (2020) [13] proposed a new method called water splitting coagulation (WSC), which synchronizes the treatment of wastewater containing both metal and organic pollutants. WSC uses water splitting in the bipolar membrane (BM) to constructively generate flocculation components (Ni (OH)(x)(2-x) +) by controlling the hydroxide transfer and cation transfer within BM and on the cation exchange membrane. Through using water cracking in BM, metal ion contaminants (M^n+^, i.e., Ni^2+^, Fe^3+^, Cr^3+^/Cr^6+^, Co^2+^) in electroplating wastewater are combined with free hydroxide ions and form a structure of controllable flocculation. Due to the water splitting in BM and the transition of metal ions on the cation exchange membrane, the water decomposition in BM and the transfer of metal ions across the cation exchange membrane is precisely controlled by adjusting the relevant parameters. Active ion migration during the WSC process follows a delivery mechanism, and it constructively results in a flocculating constituent (M(OH)_x_^(n-x)+^) by controlling the hydroxide delivery and cation delivery inside a BM and across a cation exchange membrane. Sure enough, the metal hydroxide is capable of absorbing textile dyes in (Dye)_y_M(OH)_x_^(n-x)+^ form following the interaction as that in an electro-coagulation process. Results manifesting this technology have great potential in complex industrial wastewater treatment. Membrane technology has become one of the important technologies for wastewater treatment in the printing and dyeing industry. Using literature metrology with National Knowledge Infrastructure (CNKI) and Web of Science (TM) (SCI), Liu et al. (2017) [14] studied the application status and prospect of membrane technology in wastewater treatment of printing and dyeing industry. The results showed that by 2015, the total capacity of the membrane technology in dyeing wastewater treatment in China was about 662,000 m(3).d(-1) and the number of applications was 128 (with capacity >= 500 m(3).d(-1)). Besides, “Ultrafiltration (UF) + ‘reverse osmosis’ (RO)” was the most widely applied process of membrane technologies in dyeing wastewater treatment, and the “membrane bioreactor (MBR) + RO” and “Continuous Membrane Filtration (CMF) + RO” were closely behind. Membrane technology is a promising and important technology in the wastewater treatment of the printing and dyeing industry.

The use of plants and natural processes to treat wastewater is an issue of interest to technicians and scientists around the world. Taking a southwestern sewage treatment plant in Poland as the research project, Bawiec et al. (2018) [15] analyzed the effects of temperature and sunlight on nitrate removal from hydroponic wastewater under greenhouse conditions. The findings denoted under mild climate conditions that the amount of solar radiation reaching the earth’s surface is not enough to ensure an effective year-round wastewater treatment process for hydroponic systems. Traditional wastewater treatment procedures are often insufficient to remove emerging contaminants such as PhACs (pharmaceuticals). Photocatalysis is an advanced oxidation process (AOP) that has been widely used in the removal of PhACs from wastewater due to its low operating cost. However, the problem of photocatalytic complete mineralization of PhACs is still a challenge. Based on the above background, Akpotu et al. (2019) [16] reviewed photocatalytic degradation, biodegradation, and the mechanism of degradation of phenolic compounds in wastewater and introduced the application of photocatalytic biodegradation system to degradation of PhACs in wastewater. The results deemed that a complete photocatalytic/biodegradation system is the key to complete mineralization of PhACs. Aerated wetland is an increasingly recognized natural wastewater treatment technology that relies heavily on mechanical aeration, but the relationship between volume oxygen mass transfer coefficient of wastewater in aerated wetland and organic carbon concentration remains unacquainted. Boog et al. (2020) [17] used clean water and pilot horizontal flow aerated wetland wastewater to treat domestic sewage and conducted oxygen migration experiments in laboratory-scale gravel columns. By increasing soluble CODs, the factor describing the ratio of volumetric oxygen transfer coefficient to clean water in wastewater was reduced. The derived regression equation alpha = 1.066 − 1.372 × 10^−3^ mg CODs l-1 was incorporated into the numerical process model to simulate the effect of reduced oxygen migration on the hypothetical HF aerated wetland. Simulation results revealed that a high concentration of organic carbon will reduce oxygen migration in HF aerated wetland systems, thus reducing the treatment effect. Abbasi and Tauseef (2018) [18] reported a novel plate-flow-root horizontal bioreactor (SHEFROL (R)) on their own earlier development for the first time, hinting that the use of artemisoma annua can be used to treat wastewater quickly and efficiently. In addition to extensive primary and secondary treatments in the removal of suspended solids, chemical oxygen demand, and biological oxygen demand, E. prostrata is capable of substantially removing excess nutrients and heavy metals such as copper, nickel, and manganese leading to eutrophication (nitrogen and phosphorus); the system is expected to yield significant results in sewage treatment. Using the example of Cape Cod, Massachusetts, U.S.A., Perry et al. (2020) [19] detected that biofiltration and biofiltration systems can be used to treat sewage to reduce the pollutant load in sewage pipes and receiving water because they are highly efficient at removing pollutants and can adapt to different field conditions. Retained soil filters (RSFs) for a vertically flowing constructed wetland have been successfully tested as a form of continuous post-treatment of sewage from sewage treatment plants, however, RSFs cannot be used in dry weather conditions. Given that, Brunsch et al. (2020) [20] brought up a new method that uses a double retained soil filter. In dry weather, RSFs can be used to polish sewage from sewage treatment plants, and in overflow events can help retain soil filters to treat combined sewage overflow. The study was conducted in two pilot cities, which identified dual-use RSF is a promising approach to wastewater treatment that can be expanded and employed.

The difficulty of dewatering residual sludge is the main problem of sewage treatment. Zhang et al. (2019) [21] employed chitosan (CTS), an organic polymer flocculant widely used in water and sewage treatment, in sludge treatment. After CTS treatment, the moisture content of sludge cake decreased from 85.9% to 83.0%, SV30 to about 1/2, the volume of sludge decreased to 82.9%, and the precipitation and dehydration performance of sludge were greatly improved. Abu Qdais (2019) [22] also took an in-depth look at sludge treatment by using the multi-criteria analytic hierarchy process (AHP) to build an AHP model for optimal sludge management to help Jordan’s water authorities deal with sludge from sewage treatment plants. The AHP model included three main standards, nine sub-standards, and five sludge management alternatives. The analysis implicated that the priority of the sludge management scheme is as follows: recovery of energy from sludge is the highest priority option, followed by composting, untreated disposal, and evaporation tanks, and finally the least priority option is the production of building materials from sludge.

### 2.3. Health Effects of Municipal Sewage Treatment Residues

Volker et al. (2019) [23] quantitatively evaluated in vitro (100 species) and in vivo (20 species) data, respectively. To sum up, the results demonstrated that while traditional treatment methods can effectively reduce toxicity, residual effects in wastewater may pose a risk to the ecosystem based on effect trigger values. Lopes et al. (2020) [24] detected bacterial community structure by denatuated gel gradient electrophoresis (DGGE) and evaluated antibiotic resistance genes (ARG) by polymerase chain reaction (PCR). ARG has been detected in sludge samples after alkalization treatment, which may have an impact on human health. Current technologies used in sewage treatment plants (STPs) and WWTPs do not completely eliminate pollutants such as non-steroidal anti-inflammatory drugs (NSAIDs). Almeida et al. (2020) [25] indicated NSAIDs have been found in a variety of environmental water samples, with concentrations ranging from ng/L to mu g/L, causing serious environmental and public health problems. Assress et al. (2020) [26] conducted seasonal measurements of incoming and effluent water samples from three sewage treatment plants and one drinking water treatment plant in South Africa for eight commonly used azole antifungal agents. Moreover, the risk quotient (RQ) method was used to investigate human health risks associated with wastewater and drinking water. Human health risk assessments validated that fluconazole poses a high risk in wastewater and drinking water and may cause harm to human health and safety.

Metals and chemicals in wastewater undoubtedly have special toxicity. Bozecka and Sanak-Rydlewska (2018) [27] pointed out that metals interfering with the natural biological balance and inhibiting self-cleaning processes in water have particular toxic effects, such as cobalt, which enter the environment from industrial wastewater from electrochemical plants and metallurgical industries. Supporting this notion, Alharbi and El-Sorogy (2019) [28] collected 27 samples of coastal seawater and analyzed Al, V, Cr, Mn, Fe, Co, Ni, Cu, Zn, As, Sr, Cd, Hg, and Pb using an inductively coupled plasma mass spectrometer. The results exhibited that the concentration order of bb3 is: Sr > Ni > V > Cu > Zn > Al > Fe > Cr > Mn > Pb > 0 Co > 1 Cd > 2 Hg. This proved that the harmful substances in sewage do great harm to the human body. Ma et al. (2020) [29] validated that the heavy metal particles in acid industrial wastewater seriously harm the environment and public health. The effect of pH on the detection of toxic metals in wastewater was also studied by laser-induced breakdown spectroscopy and phase transition. The findings validated that the sensitivity of heavy metal elements in acidic wastewater could be significantly improved by optimizing the pH value of libs-pt solution. Wierzbicka (2020) [30] argued that nitrates and nitrites in sewage are harmful to human health when the concentration of them exceeds the safe level. In the end, the study provided a way to measure the concentration of these compounds by using electrochemical sensors to determine nitrates, thereby reducing the human impact of nitrates and nitrites in sewage.

Many studies have demonstrated that people who frequently touch wastewater or live near sewage treatment plants are susceptible to disease. Alawi et al. (2018) [31] measured the concentration of polycyclic aromatic hydrocarbons (PAHs) in inlet, outlet, and sludge samples from five sewage treatment plants in Jordan. They found that the total concentration of PAHs in the inlet samples is 1.163~2.866g/mL, the total concentration of PAHs in the outlet samples is 0.518~1.635g/mL, and the total concentration of PAHs in the sludge samples is 2.430~5.020g/g. In the studied sludge samples, the total cancer risk of exposure to PAHs is between 3.25×10(−5) and 7.43×10(−5). In Jordan, the number of people suspected of developing cancer from exposure to sewage treatment plant sludge ranges from 33 to 74 per million. This suggests that people exposed to wastewater have an increased risk of cancer.

Dehghani et al. (2018) [32] explored the concentration of bacteria and fungi in the air at a sewage treatment plant in southwestern Iran between September 2015 and May 2016. In total, 600 samples of bacteria and fungi were collected from around the operation unit and compared spatially and seasonally, indicating that bioaerosols produced by sewage treatment plants pose a threat to the health of factory workers and nearby residents. Brisebois et al. (2018) [33] assessed the presence of 11 viral pathogens in four wastewater treatment centers (WTCs) and used a metagenomic approach to describe the viral community in the air of one WTC. The presence of viruses in WTCs’ aerosols at different locations was evaluated, and the results of four common air samplers were compared. The study examined 4 of 11 viruses, including human adenovirus, rotavirus, hepatitis a virus, and herpes simplex virus type 1. The results of metagenic analysis revealed rare viral RNA sequences in the WTC aerosol, while the sequences from human DNA viruses are relatively much richer. WTC staff may be susceptible to viral diseases such as the common cold, influenza, and gastrointestinal infections.

## 3. Research Method

Efficiency mainly describes the relationship between input and output factors. Through efficiency measurement, we can understand the performance of a group of input factors in the output process. Based on the concept of Farrell (1957) [34], Charnes et al. (1978) [35] extended his theory to establish a generalized mathematical linear programming model, called the CCR (abbreviations of Charnes, A.C.; Cooper, W.W.; Rhodes, E.L.) model, that can measure multiple inputs and multiple outputs of fixed returns to scale. In 1984, Banker et al. (1984) [36] proposed the BCC model and revised variable return to scale (VRS) assumed by the CCR model to VRS. The CCR model and the BCC model measure radial efficiency—that is, they assume that the input or output terms could increase or decrease in equal proportion. In 2001, Tone (2001) [37] proposed the difference variable model (Slacks-Based Measure, SBM), which uses the difference variable as the measurement basis, while considering the slack between input and output and presenting SBM efficiency in a non-radial estimation and scalar value.

Färe et al. (2000) [38] came up with Network Data Envelopment Analysis (Network DEA), which states that the production process is composed of many secondary production technologies, and the secondary production technologies are regarded as Sub-DMUs. Aside from these, the optimal solution is obtained by using the traditional CCR and BCC models. Compared with the traditional DEA model, these secondary production technologies are identified as “black boxes”. Moreover, the Network DEA model applies these secondary production technologies to explore the impact of input allocation and intermediate wealth on the production process. Following Färe et al., Tone and Tsutsui (2009) [39] put forward the weighted SBM Network DEA model, whereby the linkage among various departments of the decision-making unit is taken as the analysis basis of the Network DEA model, and each department is regarded as a Sub-DMU. In the network DEA model, a dynamic approach is allowed, in which the DMU is evaluated at different time periods and cargos are introduced to connect the stages that make up the DMU in different periods (Tone and Tsutsui (2010) [40]). Dynamic DEA has developed because Kloop (1985) [41] proposed Window analysis in 1985. Using the dynamic analysis model in the first place, Färe and Grosskopf (1996) [42] were the first to put interlinked activities into dynamic analysis, with Kao and Hwang (2008) [43], Nemoto and Goto (1999, 2003) [44,45], Chang et al. (2009) [46], and other scholars publishing relevant analysis models successively.

Tone and Tsutsui (2014) [47] proposed the weighted SBM Dynamic Network DEA model with the linkage among various departments of the decision-making unit taken as the analysis basis of the Network DEA model and each department regarded as a Sub-DMU. Carryover activities are taken as the linkage, but Tone and Tsutsui’s dynamic network DEA model does not consider undesirable output. Because the dynamic network DEA model does not consider undesirable factors, in order to solve the problem of the undesirable factors and a multi-stage process, this paper proposes a modified two-stage dynamic data envelopment analysis model that combines the dynamic network DEA model and undesirable factors in order to evaluate the two stages of China’s urban sewage treatment and health from 2014–2017. The target is to avoid an underestimation or overestimation of efficiency value and improvement.

### 3.1. Modified Two-Stage Dynamic Data Envelopment Analysis Model 

Suppose there are *n*
*DMU*s (*j* = 1,…,*n*), with each having *k* divisions (*k* = 1,…,*K*), and *T* time periods (*t* = 1,…,*T*). Each *DMU* has an input and output at time period *t* and a carryover (link) to the next *t*+1 time period.

Set *m_k_* and *r_k_* to represent the inputs and outputs in each division *K*, with (*k,h*)*i* representing divisions *k* to *h* and *L_hk_* being the *k* and *h* division set. The inputs, outputs, links, and carryover definitions are outlined in the following paragraphs.

#### 3.1.1. Wastewater Treatment Stage

$X_{1}^{t}$: Sewage treatment plants as input.

$Y_{1good}^{t}$: Total wastewater discharge.

$Y_{1bad}^{t}$: Municipal sewage treatment capacity and COD concentration.

$Z_{(12)in}^{t}$(link between wastewater treatment stage and health stage): Heavy metal pollutant equivalent concentration.

#### 3.1.2. Health Stage

$X_{2}^{t}$: Number of health technicians as input and local fiscal medical and health expenditure as input.

$Y_{2good}^{t}$: Average life expectancy.

$Y_{2bad}^{t}$: Carcinogenic risk.

$Z_{ok_{l}input}^{\left(t,(t+1)\right)}$(Carryover): Completed investments in wastewater treatment projects.

The following is the non-oriented model:

(a) Objective function
Overall efficiency:$\theta_{0}^{*}=\min\frac{\sum_{t=1}^{T}W^{t}\left[\sum_{k=1}^{K}W^{k}\left[1-\frac{1}{m_{k}+linkin_{k}+ninput_{k}}(\sum_{i=1}^{m_{k}}\frac{S_{iok}^{t-}}{x_{iok}^{t}}+\sum_{{\left(kh\right)}_{l}=1}^{linkin_{k}}\frac{s_{o{\left(kh\right)}_{l}in}^{t}}{z_{o{\left(kh\right)}_{l}in}^{t}}+\sum_{k_{l}}^{ninput_{k}}\frac{s_{ok_{l}input}^{\left(t,t+1\right)}}{z_{ok_{l}input}^{\left(t,t+1\right)}})\right]\right]}{\sum_{t=1}^{T}W^{t}\left[\sum_{k=1}^{K}W^{k}\left[1+\frac{1}{r_{1k}+r_{2k}}(\sum_{r=1}^{r_{1k}}\frac{s_{rokgood}^{t+}}{y_{rokgood}^{t}}+\sum_{r=1}^{r_{2k}}\frac{s_{rokbad}^{t-}}{y_{rokbad}^{t}})\right]\right]}$Subject to:Production stage$x_{o1}^{t}=X_{1}^{t}\lambda_{1}^{t}+s_{1o}^{t-}(\forall t)\;;$$y_{o1good}^{t}=Y_{1good}^{t}\lambda_{1}^{t}-s_{1ogood}^{t+}\left(\forall t\right)$$y_{o1bad}^{t}=Y_{1bad}^{t}\lambda_{1}^{t}+s_{1obad}^{t-}\left(\forall t\right)$$\lambda_{1}^{t}\geq 0,s_{1o}^{t-}\geq 0,s_{1ogood}^{t+}\geq 0,\left(\forall t\right)$$Z_{o(12)in}^{t}=Z_{(12)in}^{t}\lambda_{1}^{t}+S_{o(12)in}^{t}\left((1,2)in\right)$Health stage$x_{o2}^{t}=X_{2}^{t}\lambda_{2}^{t}+s_{2o}^{t-}\left(\forall t\right)$$y_{o2good}^{t}=Y_{2good}^{t}\lambda_{2}^{t}-s_{2ogood}^{t+}\left(\forall t\right)$$y_{o2bad}^{t}=Y_{2bad}^{t}\lambda_{2}^{t}+s_{2obad}^{t-}\left(\forall t\right)$$\lambda_{2}^{t}\geq 0,s_{2o}^{t-}\geq 0,s_{2ogood}^{t+}\geq 0,s_{2obad}^{t-}\geq 0,\left(\forall t\right)$$e\lambda_{k}^{t}=1_{\left(\forall k,\forall t\right)}$$Z_{o(kh)in}^{t}=Z_{(kh)in}^{t}\lambda_{k}^{t}+S_{o(kh)in}^{t}\left((kh)in=1,\ldots ,linkin_{k}\right)\sum_{j=1}^{n}z_{jk_{1}\alpha }^{\left(t,(t+1)\right)}\lambda_{jk}^{t}=\sum_{j=1}^{n}z_{jk_{1}\alpha }^{\left(t,(t+1)\right)}\lambda_{jk}^{t+1}$$\left(\forall k;\forall k_{l};t=1,\ldots ,T-1\right)$(1)$$Z_{ok_{l}input}^{\left(t,(t+1)\right)}={\sum_{j=1}^{n}z}_{jk_{l}input}^{\left(t,(t+1)\right)}\lambda_{jk}^{t}+s_{ok_{l}input}^{\left(t,(t+1)\right)}k_{l}=1,\ldots ,ngood_{k};\forall k;\forall t)$$

(b) Period and division efficiencies
(b1) Period efficiency:(2)$$\partial_{0}^{*}=\min\frac{\sum_{k=1}^{K}W^{k}\left[1-\frac{1}{m_{k}+linkin_{k}+ngood_{k}}(\sum_{i=1}^{m_{k}}\frac{S_{iok}^{t-}}{x_{iok}^{t}}+\sum_{{\left(kh\right)}_{l}=1}^{linkin_{k}}\frac{s_{o{\left(kh\right)}_{l}in}^{t}}{z_{o{\left(kh\right)}_{l}in}^{t}}+\sum_{k_{l}}^{ngood_{k}}\frac{s_{ok_{l}good}^{\left(t,t+1\right)}}{z_{ok_{l}good}^{\left(t,t+1\right)}})\right]}{\sum_{k=1}^{K}W^{k}\left[1+\frac{1}{r_{1k}+r_{2k}}(\sum_{r=1}^{r_{1k}}\frac{s_{rokgood}^{t+}}{y_{rokgood}^{t}}+\sum_{r=1}^{r_{2k}}\frac{s_{rokbad}^{t-}}{y_{rokbad}^{t}})\right]}$$(b2) Division efficiency:(3)$$\phi_{0}^{*}=\min\frac{\sum_{t=1}^{T}W^{t}\left[1-\frac{1}{m_{k}+linkin_{k}+ninput_{k}}(\sum_{i=1}^{m_{k}}\frac{S_{iok}^{t-}}{x_{iok}^{t}}+\sum_{{\left(kh\right)}_{l}=1}^{linkin_{k}}\frac{s_{o{\left(kh\right)}_{l}in}^{t}}{z_{o{\left(kh\right)}_{l}in}^{t}}+\sum_{k_{l}}^{ninput_{k}}\frac{s_{ok_{l}input}^{\left(t,t+1\right)}}{z_{ok_{l}input}^{\left(t,t+1\right)}})\right]}{\sum_{t=1}^{T}W^{t}\;\left[1+\frac{1}{r_{1k}+r_{2k}}(\sum_{r=1}^{r_{1k}}\frac{s_{rokgood}^{t+}}{y_{rokgood}^{t}}+\sum_{r=1}^{r_{2k}}\frac{s_{rokbad}^{t-}}{y_{rokbad}^{t}})\right]}$$(b3) Division period efficiency:(4)$$\rho_{0}^{*}=\min\frac{1-\frac{1}{m_{k}+linkin_{k}+ninput_{k}}(\sum_{i=1}^{m_{k}}\frac{S_{iok}^{t-}}{x_{iok}^{t}}+\sum_{{\left(kh\right)}_{l}=1}^{linkin_{k}}\frac{s_{o{\left(kh\right)}_{l}in}^{t}}{z_{o{\left(kh\right)}_{l}in}^{t}}\sum_{k_{l}}^{ninput_{k}}\frac{s_{ok_{linput}input}^{\left(t,t+1\right)}}{z_{ok_{l}input}^{\left(t,t+1\right)}})}{1+\frac{1}{r_{1k}+r_{2k}}(\sum_{r=1}^{r_{1k}}\frac{s_{rokgood}^{t+}}{y_{rokgood}^{t}}+\sum_{r=1}^{r_{2k}}\frac{s_{rokbad}^{t-}}{y_{rokbad}^{t}}+)}$$

### 3.2. Input, Desirable Output, and Undesirable Output Efficiency

Hu and Wang’s (2006) [48] total-factor energy efficiency index can be used to overcome any possible biases in the traditional energy efficiency indicators, for which there are eleven key efficiency models here in this present study: sewage treatment plants as input, total wastewater discharge, municipal sewage treatment capacity, municipal sewage treatment capacity, COD concentration, heavy metal pollutant equivalent concentration, number of health technicians as input, local fiscal medical and health expenditure as input, average life expectancy, carcinogenic risk, and investment in fixed assets.

The efficiency models are defined as formula (5)–(7):(5)$$Input\;efficiency=\frac{Target\;input}{Actual\;input}$$
(6)$$Undesirable\;output\;efficiency=\frac{Target\;Undesirable\;output}{Actual\;Undesirable\;output}$$
(7)$$Desirable\;output\;efficiency=\frac{Target\;Desirable\;output}{Actual\;Desirable\;output}$$

If the target inputs equal the actual inputs, then the efficiencies are 1, which indicates overall efficiency; however, if the target inputs are less than the actual inputs, then the efficiencies are less than 1, which indicates overall inefficiency.

If the target desirable outputs are equal to the actual desirable outputs, then the efficiencies are 1, indicating overall efficiency; however, if the target desirable outputs are more than the actual desirable outputs, then the efficiencies are less than 1, indicating overall inefficiency.

If the target undesirable outputs are equal to the actual undesirable outputs, then the efficiencies are 1, indicating overall efficiency; however, if the target undesirable outputs are less than the actual undesirable outputs, then the efficiencies are less than 1, indicating overall inefficiency.

## 4. Empirical Analysis

### 4.1. Data Description

#### 4.1.1. Explanation of Variables

This paper evaluates the wastewater treatment efficiency and health efficiency of 30 provincial administrative units based on the two-stage dynamic DEA model. As the focus of the study is on the provinces in China, Taiwan and Hong Kong and Macao special administrative regions are not analyzed. In addition, due to limited data of Tibet autonomous region, it is also not included.

In the wastewater treatment stage, completed investment in wastewater treatment project and sewage treatment plants are adopted as the input variables. Municipal sewage treatment capacity is the desirable output, while total wastewater discharge, COD concentration, and heavy metal pollutant equivalent concentration are undesirable output variables. Among them, completed investment in wastewater treatment project is selected as the carryover indicator, and heavy metal pollutant equivalent concentration is an intermediate variable. In the health stage, number of health technicians and local fiscal medical and health expenditure are taken as input variables. Average life expectancy and carcinogenic risk are agreed and not agreed outputs, respectively. See Table 1 for details.

The data on completed investment in wastewater treatment project, total wastewater discharge, COD concentration, number of health technicians, local fiscal medical and health expenditure, and average life expectancy are from the provincial annual data of the National Bureau of Statistics from 2014 to 2017. Data on sewage treatment plants and municipal sewage treatment capacity are obtained from China Environmental Statistics Yearbook 2014–2017. Heavy metal pollutant equivalent concentration and carcinogenic risk are calculated on the basis of different heavy metal concentrations from China Environmental Statistics Yearbook. The specific variables are described as follows.

① Completed investment in wastewater treatment project (investment). It refers to the investment that has been completed in a project to treat wastewater.

② Sewage treatment plants. It refers to the number of sewage treatment plants in a province (municipality directly under the central government, autonomous region).

③ Total wastewater discharge. It refers to the sum of industrial wastewater discharge and domestic sewage discharge.

④ Municipal sewage treatment capacity. It is defined as the total amount of sewage treated in a province (municipality directly under the central government, autonomous region) in a year.

⑤ COD concentration. It is defined as the concentration of oxygen required to oxidize organic pollutants in water with chemical oxidants. COD refers to the use of chemical oxidants (such as potassium dichromate) in water reducing substances (such as organic matter) and the oxidation decomposition of oxygen consumption, reflecting the extent of water pollution by reducing substances. The reducing substances can reduce the content of dissolved oxygen in the water, leading to the death of organisms in the water due to hypoxia and the deterioration of water quality. A higher COD denotes a higher content of reducing substances in the water and the more serious pollution. Since organic matter is the most common reducing substance in water, COD is an important parameter to measure organic pollution.

⑥ Heavy metal pollutant equivalent concentration. It is calculated on the basis of different heavy metal concentrations from China Environmental Statistics Yearbook. It refers to the degree of harm to the environment. The higher the equivalent concentration of pollution is, the greater is the degree of harm to the environment. According to China’s environmental quality standard for surface water GB3838-2002, the heavy metal index includes 6 items: cadmium (Cd), lead (Pb), chromium (Cr), nickel (Ni), zinc (Zn) and copper (Cu). However, there are some essential elements to support life, such as Zn, Cu and so on. No matter the lack or surplus of these elements, they will affect human health. There are other heavy metal elements, such as cadmium, chromium, etc., which have obvious toxic effects. No matter how they get into the body, they will cause poisoning, leading to serious illness and even death. Based on the existing literature [49,50] and the quality monitoring data of Dalian’s key drinking water sources [51], chemical carcinogens include hexavalent chromium, cadmium and arsenic. So in this paper, hexavalent chromium, cadmium and arsenic are used as the indexes affecting health.

⑦ Number of health technicians. Health Technicians includes practicing doctors, assistant practicing doctors, registered nurses, pharmacists (judges), test technicians (judges), image and trainee medical technicians, hygiene supervisors (medicine, nursing, skills) and other health professionals.

⑧ Local fiscal medical and health expenditure. It refers to the medical and health expenditure items in the general budget of the local government. It includes expenditure on medical and health management affairs, expenditure on medical services, expenditure on medical security, expenditure on disease prevention and control, expenditure on health supervision, expenditure on maternal and child health care, expenditure on rural health, etc.

⑨ Average life expectancy. It refers to the number of years that people can continue to live after the exact age of X at a certain age-specific mortality level. It is an indicator to measure the health level of residents in a country, a nation, or a region and can reflect the quality of life in a society.

⑩ Carcinogenic risk. It calculates the carcinogenic risk value of total chromium emission, arsenic emission, and cadmium emission. The health risks of individual carcinogenic pollutants in multiple exposure pathways are as follows:(8)$$\left\{ \begin{array}{ll} R_{i}=CDI\times Sf, & R<0.01\\ R_{i}=1-\exp\left[-\left(CDI\times Sf\right)\right], & R>0.01 \end{array}\right.$$

In the formula, Ri represents the health risk value of a single pollutant under various exposure pathways, CDI represents the exposure dose, Sf represents the carcinogenic slope factor of the pollutant, and the unit is mg·kg^−1^·d^−1^. The higher the R_i_ value is, the greater is the health risk of a carcinogen—that is, the higher the cancer probability of the pollutant. In concrete analysis, the maximum acceptable risk level of the International Council on Cancer (ICRP), 5×10^−5^, is usually taken as a reference value, which is interpreted as no more than five people per 10,000 are affected by the chemical with a new disease or cancer. The formula for calculating the total risk of various carcinogens is shown below.
(9)$$R_{i}^{t}=\sum_{i=1}^{j}R_{i}$$
Here, $R_{i}^{t}$ represents the total health risk of all pollutants in all exposure pathways.

Figure 1 illustrates the flow structure of this paper by using a flow chart. See Figure 1 for details.

#### 4.1.2. Data Description

This study selects the input and output data of 30 provinces in China from 2014 to 2017 to calculate the average, the maximum, the minimum, and the standard values of completed investment in wastewater, treatment project, sewage treatment plants, total wastewater discharge, municipal sewage, treatment capacity, COD concentration, heavy metal pollutant equivalent concentration, number of health technicians, local fiscal medical and health expenditure, average life expectancy, and carcinogenic risk. See Table 2 for details.

### 4.2. Overall Efficiency Analysis

This section calculates the overall efficiency of each province from 2014 to 2017 and ranks the 30 provinces in descending order according to their overall efficiency. From 2014 to 2017, the total efficiency values of DEA in the two stages from wastewater treatment input to health output of 30 provinces in China reveal that the overall efficiency of Ningxia and Qinghai is 1 for all four years, reaching the optimal state. See Table 3 for details.

The total efficiency of Hunan is 0.6261 in 2014, 0.7270 in 2015, and 1 in both 2016 and 2017, meaning the resource utilization efficiency is at the optimal state. On the contrary, Hainan, where the overall efficiency of the four years is the third highest, has an efficiency value of 1 in 2014 and 2015, but then the total efficiency value of the following two years falls to 0.6728 and 0.6566, indicating a deterioration of resource integration there. In total, the overall efficiencies of Gansu, Jiangsu, Xinjiang, Anhui, and Henan advance steadily in these four years, while Inner Mongolia displays a slow decline, and the overall efficiency of Fujian plummets to 0.3519 in 2017.

The highest value of overall efficiency for many provinces appears in 2016, such as Liaoning, Zhejiang, Heilongjiang, Shanxi, and Chongqing. The total efficiencies of Tianjin and Beijing increase steadily in the first three years and reach 1 in 2016, but then these two municipalities directly under central government control plummet to approximately 0.6 in 2017 and fail to maintain an optimal state. The highest value of Guangxi’s overall efficiency is 0.8034 in 2015, and then its overall efficiency in the other three years is about 0.4. The efficiency values of Shanghai, Guangdong, Yunnan, and Hubei change little in these four years, while those of Guizhou, Jilin, Shandong, Shaanxi, Sichuan, and Hebei change slightly, but their overall efficiency values are still at a low level.

Figure 1 compares the distribution of total efficiency in the 30 provinces from 2014 to 2017. The gap in total efficiency can be clearly seen through the radar chart. See Figure 2 for details.

Figure 3 shows the geographical distribution of the overall efficiency of 30 provinces from 2014 to 2017. See Figure 3 for details.

### 4.3. Efficiency Comparison of the Two Stages

The efficiency of the wastewater treatment stage is visibly higher than that of the health stage, and many provinces reach the optimal state in the first stage. For example, the efficiencies of the wastewater treatment stage of Beijing, Guangdong, Hunan, and Shanghai are 1 from 2014 to 2017, and the efficiency values of Fujian and Gansu are 1 for three consecutive years. On the whole, the two stages illustrate a steady but slow growth trend, indicating that the five development concepts of “innovation, coordination, green development, openness, and sharing” have been deeply rooted in the hearts of the country’s citizens. As for the wastewater treatment stage, the efficiencies of Beijing, Guangdong, Hunan, Qinghai, Ningxia, and Shanghai are 1 from 2014 to 2017, and those of Gansu, Fujian, Inner Mongolia, and Zhejiang reach 1 for three years. However, there are still many provinces with low efficiency values. Guizhou, Hebei, Jilin, and Chongqing all have efficiency values below 0.5 in the four years. These provinces should take sewage treatment into account and make the best use of capital and personnel. See Table 4 for details.

The efficiency of the wastewater treatment stage has an obvious promoting effect on the total efficiency of each province, while the health stage to some extent inhibits the continuous growth of the total efficiency value of each province. For the four years, the efficiency of wastewater treatment in Beijing is 1. However, since the efficiency value of the second stage reaches an optimal state only in 2016, while it is around 0.3 in the other three years, bringing the total efficiency of Beijing to around 0.7 and ranking seventh in China. The efficiency of wastewater treatment in Guangdong is 1 for the four years, and that of the health stage is 0.1630, 0.1731, 0.1305, and 0.2297 from 2014 to 2017. The total efficiency is about 0.6, indicating that the health stage clearly is below total efficiency.

The efficiency of each province is closely related to geographical location, economic development, government policies, and other factors. The efficiencies of Ningxia and Qinghai in the two stages from 2014 to 2017 are 1, which is the top in China, thanks to their superior geographical location and the implementation of environmental protection concepts as well as due to the small number of factories and economic backwardness there. For Hunan and Shanghai, their efficiencies of wastewater treatment are 1 in each of the four years because of their developed economies and advanced wastewater treatment equipment. In these four years, the efficiencies of the two stages for Shaanxi and Chongqing are relatively low, rarely exceeding 0.5. In 2015, the efficiency of the health stage in Shaanxi is only 0.1531, or far behind other provinces. Both Shaanxi and Chongqing are heavily industrialized cities with severe pollution and have poor environmental protection awareness. Therefore, they must balance the relationship between economic development and environmental protection.

The average efficiency of the wastewater treatment stage is 0.6837, and that of the health stage is 0.4243 by calculation. We observe that the efficiency of the first stage is obviously higher than that of the second stage. Based on the average efficiency of each province, we divide the studied areas into four parts: high-high, low-low, high-low, and low-high. Among them, eight provinces including Beijing, Gansu, Hunan, Liaoning, Inner Mongolia, Ningxia, Qinghai, and Tianjin have higher values than the average efficiency in the two stages, while ten provinces including Hebei, Henan, Hubei, Jilin, Shandong, Shaanxi, Sichuan, Chongqing, Xinjiang, and Guizhou have lower values than the average efficiency in the two stages. Anhui, Guangxi, Hainan, Heilongjiang, Jiangxi, and Shanxi have efficiencies in the second stage that are higher than the average level, but their efficiencies in the first stage are lower than the average level. Fujian, Guangdong, Jiangsu, Shanghai, Yunnan, and Zhejiang have higher efficiencies than the average level in the first stage, but lower than average efficiencies in the second stage. Therefore, the health stage needs great improvement. See Figure 4 for details.

### 4.4. Itemized Efficiency Analysis

#### 4.4.1. Sewage Treatment Plants’ Efficiency Analysis

The efficiency value of many provinces reflects a trend of steadily increasing, with Gansu rising from 0.8586 in 2014 to 1 in 2015, 1 in 2016, and 1 in 2017. Jiangsu goes from 0.4809 in 2014 to 0.6838 in 2015, reaching 1 in both 2016 and 2017. Beijing, Guangdong, Hunan, Ningxia, Qinghai, and Shanghai all have an efficiency value of 1 for the four years, while Gansu, Zhejiang, Fujian, and Inner Mongolia have an efficiency value of 1 for three consecutive years. We see that these provinces attach great importance to the sewage treatment problem and have invested manpower, material resources, and financial resources to treat sewage and achieve outstanding results. However, in some provinces, the efficiencies do not increase significantly or even decline. The efficiency value of Guizhou is at a low level of 0.2–0.4. Shandong has a small range of 0.4–0.5. Jilin has a four-year efficiency value of about 0.5. Guangxi decreases from 0.9211 in 2014 to 0.7581 in 2017. Heilongjiang decreases from 0.6638 in 2014 to 0.3511 in 2017. See Table 5 for details.

#### 4.4.2. Total Wastewater Discharge Efficiency Analysis

Beijing, Gansu, Guangdong, Hunan, Ningxia, Qinghai, and Shanghai have total wastewater discharge efficiencies of 1 for all four years. The efficiencies of Inner Mongolia, Fujian, and Zhejiang for 2014–2016 are 1. The efficiencies of Jiangsu, Tianjin, Heilongjiang, and Hainan are 1 for two consecutive years. Nonetheless, this efficiency variable generally presents a slight downward trend. The efficiencies of Inner Mongolia, Fujian, and Zhejiang in the first three years are 1, but then drop to 0.8934, 0.8082, and 0.6339 in 2017, respectively. Tianjin falls from 0.9004 in 2014 to 0.7884 in 2017, or down by 0.1120. Xinjiang falls from 0.8954 in 2014 to 0.6342 in 2017, or down by 0.2612. Hainan owns the biggest drop from 1 in 2014 and 2015 to 0.5048 in 2014, or down by 0.4952. Guizhou and Henan exhibit a slight change, fluctuating between 0.4 and 0.5 and ranking lower in efficiency. See Table 6 for details.

#### 4.4.3. COD Concentration Efficiency Analysis

The efficiency value of the COD concentration variable is relatively high, reaching 1 in about 10% of the provinces every year, but showing a downward trend. Fujian drops from 1 in 2014 to 0.4024 in 2017, or down 0.5976; Hainan falls by 0.9411 from 1 in 2014 to 0.0589 in 2017, and Jiangsu decreases by 0.8115 from 0.9882 in 2014 to 0.1767 in 2017. The situation is improving, and the pollutants in the water gradually decrease. All provinces should still attach great importance to the harmful substances in the water to the human body and strengthen scientific and technological investment or introduce professional equipment to degrade harmful substances in water. See Table 7 for details.

#### 4.4.4. Number of Health Technicians’ Efficiency Analysis

The efficiency values in the four years for Hainan, Ningxia, and Qinghai are 1, reaching the optimal state. Numerous provinces register their highest efficiency in 2016, including Heilongjiang, Beijing, Chongqing, Shanghai, Shanxi, and Liaoning, while those hitting their lowest are Anhui, Henan, Guangxi, Inner Mongolia, Gansu, Jiangsu, and Guangdong. The efficiency values of most provinces decrease, including Fujian, Jiangxi, Hubei, Shandong, and Liaoning, which fall significantly from 0.8530, 0.7321, 0.7001, 0.5785, and 0.6105 in 2014 to 0.3135, 0.2042, 0.1299, 0.1001, and 0.2188 in 2017, respectively. A few provinces see slow or no distinct changes in efficiency. The efficiency values of Anhui and Henan are 0.9169 and 0.7007 in 2014, but they plunge in 2015 and 2016. Anhui drops to 0.6647 in 2015 and to 0.3336 in 2016, while Henan drops to 0.5977 in 2015 and to 0.1168 in 2016. In 2017, both provinces increase by 0.0831 and 0.2993, respectively. The efficiency of Hebei in the four years is about 0.1, while Shaanxi’s efficiency is about 0.2, with little change and always lower than the national average. See Table 8 for details.

#### 4.4.5. Local Fiscal Medical and Health Expenditure Efficiency Analysis

The efficiency values of Hainan, Hunan, Ningxia, Qinghai, and Zhejiang in the four years are 1, and about 10% of the provinces reach the optimal state every year. This expenditure reveals a slow increasing trend. For example, Tianjin rises from 0.5638 in 2014 and 0.4998 in 2016 to 1 in 2016 and 2017, Heilongjiang increases from 0.3731 in 2014 to 0.5918 in 2017, and Xinjiang goes from 0.3322 in 2014 to 0.5897 in 2017. There is still great improvement in this variable. See Table 9 for details.

#### 4.4.6. Average Life Expectancy Efficiency Analysis

We note that the efficiency value of average life expectancy increases rapidly. Anhui, Beijing, Gansu, and Fujian rise to 1 in 2017 from 0.1921, 0.8226, 0.3367, and 0.2326 in 2014, respectively, increasing by 0.8079, 0.1774, 0.6633, and 0.7674. By the end of 2017, 26 provinces reach the optimal state. Guizhou, Hainan, Jilin, Heilongjiang, Ningxia, Qinghai, Shaanxi, Shanghai, Tianjin, Xinjiang, Chongqing, and other provinces all have an efficiency value of 1 in the four years, which hints that national health awareness has been enhanced and the happiness of urban residents has been improved. See Table 10 for details.

#### 4.4.7. Carcinogenic Risk Efficiency Analysis

The efficiency value of the carcinogenic risk variable decreases on the whole. Jiangxi, Shanxi, Guizhou, and Heilongjiang decrease from 1 in 2014 to 0.3264, 0.3023, 0.5330, and 0.8218 in 2017, respectively, by falling in a range of 0.6736, 0.6977, 0.467, and 0.1782. Fujian decreases from 0.7877 in 2014 to 0.3487 in 2017, Hebei from 0.3306 in 2014 to 0.1492 in 2017, Hubei from 0.6554 in 2014 to 0.4459 in 2017, Jilin from 0.9362 in 2014 to 0.2030 in 2017, and Sichuan from 0.9168 in 2014 to 0.4633 in 2017. Among them, Jiangxi, Shanxi, and Jilin have a relatively large decline of about 0.7. To conclude, the medical treatment level has been enhanced. See Table 11 for details.

## 5. Conclusions

According to the two-stage (wastewater treatment stage and health stage) dynamic SBM DEA model, this research analyzes the input and output efficiencies of 30 provinces in China, obtaining the following conclusions.

(1) The efficiency values of each province in China are influenced by geographical location, urban development, and pillar industries of each region’s economy. Ningxia, Qinghai, Beijing, and Hainan have higher efficiency values of various indicators that are close to or at the optimal state and are among the top in China. Located in the northwest inland arid region, Ningxia’s water environmental problems come mainly from agricultural water pollution, soil erosion, and water supply and demand imbalances, while its urban industrial and living wastewater is not serious. Moreover, the development of Ningxia’s urbanization is unbalanced with a smaller population and less domestic sewage, and so its efficiency is higher. Qinghai is located in the northeast of the Qinghai-Tibet Plateau, and due to its remote geographical location, its population is sparse. In addition, its economy is dominated by agriculture and animal husbandry, and so urban sewage is less. As the capital of China, Beijing is the political center, cultural center, and scientific research center, which is not based on the development of industry. Hainan is located in the southernmost part of China. Its economy is dominated by tourism, housing industry, agriculture, and low-carbon manufacturing industry. It also has a small resident population, and so it has less urban industrial wastewater and domestic sewage. Sichuan, Chongqing, Hebei, Shandong, Guizhou, and Shaanxi have lower efficiency values because the cities of Deyang and Panzhihua in Sichuan, Jinan, Weifang, and Zibo in Shandong, Handan and Tangshan in Hebei, Liupanshui in Guizhou, and Baoji in Shaanxi are all famous heavy industry cities with extremely serious industrial water pollution. Sichuan has a basin topography, Chongqing is mountainous, Guizhou is located in the southwest hinterland, and Hebei, Shandong, and Shaanxi are located in north China. Therefore, the pollutants are not easy to diffuse, and thus, the provinces mentioned above have low efficiency values.

(2) The efficiency value in the health stage is distinctly lower than that in the wastewater treatment stage, which puts a drag on the total efficiency of each province, and so there is more room to enhance efficiency. In the health stage, the efficiency of number of health technicians is significantly lower (by 0.2) than that of local fiscal medical and health expenditure. The efficiency values of number of health technicians in Hebei, Shaanxi, Jilin, Shanghai, and Xinjiang are relatively low at less than 0.5 in the four years. Shaanxi, Jilin, and Xinjiang have low efficiency values because of their remote geographical location and the gap between remuneration and workers’ treatment to the more developed areas of China. Hebei has low efficiency because of the siphon effect, thus presenting that high-quality resources are greatly concentrated in Beijing and Tianjin. Conversely, Shanghai has low efficiency values because of fierce competition and insufficient government input. The efficiency of carcinogenic risk is higher than the efficiency of average life expectancy, but the gap is narrowing. It means that carcinogenic risk efficiency is declining year by year and average life expectancy is increasing because 26 provinces in 2017 are at a level of 1, reflecting improved medical levels and the enhancement of national health consciousness. We can see that improving the efficiency of health stage mainly helps the efficiency of number of health technicians. One problem that every province should overcome is how to retain talents and give full play to the advantages of those talents.

(3) Each province should choose the best economic development mode according to its own situation to pursue a balance between economic development and environmental protection. Cities in Ningxia and Qinghai have relatively light water pollution, but the economic development of these two provinces is relatively backward. They can thus combine the original pillar industries, agriculture and animal husbandry, with “Internet +” to monitor the growth of crops or animals through artificial intelligence in real time. They may also consider simultaneously using the Internet to promote products more efficiently and cheaply, thus helping to boost sales and accelerate the development of their digital economy. At the same time, Ningxia and Qinghai could set up policies to attract investment (except for projects with high energy consumption and high pollution) and develop their own brand of special tourism. Sichuan, Chongqing, Hebei, Shandong, Guizhou, and Shaanxi should optimize their industrial structure. First, in response to the national call for mass entrepreneurship and innovation, they must gradually abandon heavy industry and develop high-tech enterprises to alleviate environmental problems such as water pollution. In addition, they can vigorously develop tourism and other service industries and go deeper into the excavation of the regional characteristics of specific investment projects. For example, Zunyi in Guizhou, Yan’an in Shaanxi, and Baiyang Lake in Hebei can develop the red tourism (taking the memorial sites and markers formed by the great achievements made by the people under the leadership of the communist party of China in the period of revolution and war as the carrier, and taking the revolutionary history, revolutionary deeds and revolutionary spirit as the connotation, we organize thematic tourism activities to remember and learn revolutionary martyr) industry and promote revolutionary traditional education.

(4) Provinces should retain health professionals in order to maintain the health of their citizens. For example, Hebei, Shaanxi, Jilin, and Xinjiang should establish a talent incentive model to improve the salary and welfare of health technicians, so that they are more willing to stay in their hometown and make contributions to medical and health care. Furthermore, each province could attract academic medical personnel to obtain employment and feasibly improve the level of local medical practices. In first-tier cities, like Shanghai, they should put people first, provide more jobs for health technicians, reduce the intensity of competition for jobs, and improve the happiness and sense of belonging of health technicians from all aspects so as to give take advantage of the personnel team.

(5) All provinces should place great importance to sewage treatment. First, enterprises and governments should target to increase technological input and introduce advanced sewage treatment equipment from abroad, or develop high-tech products independently, which would be beneficial for reaching the target of reducing the harm from chromium, arsenic, cadmium, and other substances in sewage to the human body. Second, another option is to set up efficient sewage treatment plants to prevent the secondary harm of sewage to humans and promote the recycling of water resources. Economically developed provinces such as Beijing, Shanghai, Guangdong, Zhejiang, and Jiangsu should make the best use of their economic, geographical, and talent advantages and take the lead in developing fruitful sewage treatment plants. Third, backed by national enforcement, laws and regulations should be enacted to curb the arbitrary discharge of urban production and domestic sewage and to reduce the quantity of sewage at the source. Finally, governments can strengthen environmental protection education, raises people’s environmental protection consciousness, and allow people to participate in social supervision.

(6) The central government can promote coordinated regional development. For instance, in terms of coordinated development in the Beijing-Tianjin-Hebei region, Beijing should gradually relieve itself of non-capital functions, optimize the urban layout, and expand the ecological space of environmental capacity. Hebei and Tianjin then can actively undertake the non-capital functions of Beijing, such as transforming Hebei from heavy industry to green coordinated development and initiating high-quality development of Tianjin’s economy. Coordinated development in the Yangtze River Delta can give full play to the leading role of Shanghai by sharing sewage treatment experience and technology with other cities. Jiangsu, Zhejiang, and Anhui should accept and actively learn advanced technology and give full play to their respective advantages, so as to achieve the goal of narrowing their economic development gap and to set up rational industrial division and green and sustainable economic development in the Yangtze River Delta. All provinces in China deserve to speed up the flow of factors, narrow the economic gap between regions, and finally, realize common prosperity.

The data of urban sewage treatment from 2014 to 2017 are selected in this paper. The research period is relatively short and the situation of sewage treatment in rural China isn’t taken into account. We will continue to follow up China’s sewage treatment situation in the following period.

## Figures and Tables

**Figure 1 healthcare-08-00119-f001:**
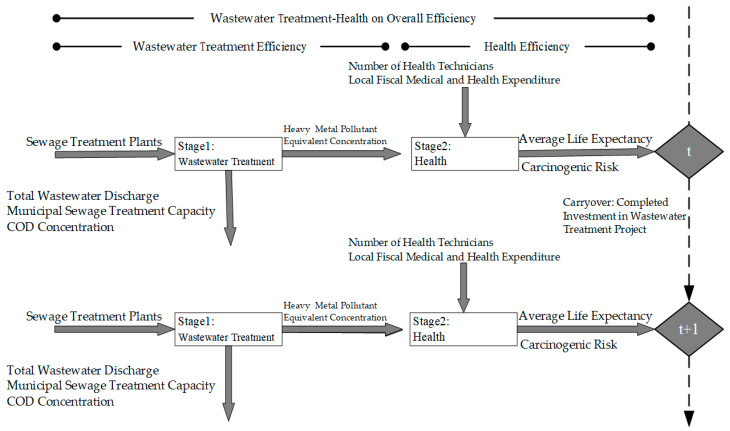
Network Dynamic Model.

**Figure 2 healthcare-08-00119-f002:**
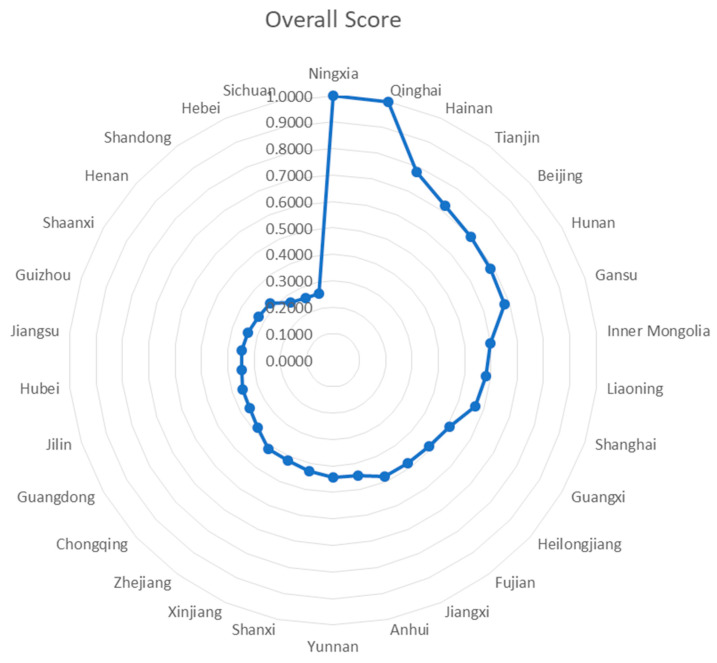
Four-year Overall Score and Total Efficiency of Provinces.

**Figure 3 healthcare-08-00119-f003:**
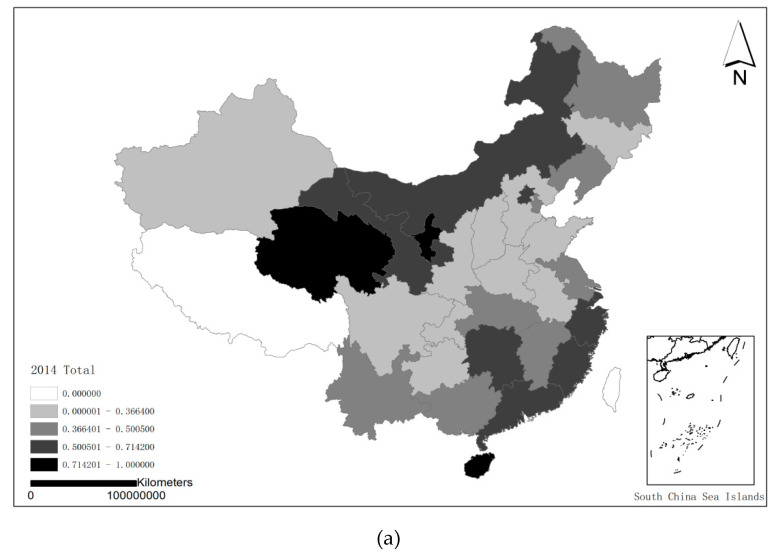
Geographical Distribution of Overall Score of Each Province in the 4 years. (**a**): 2014; (**b**): 2015; (**c**): 2016; (**d**): 2017.

**Figure 4 healthcare-08-00119-f004:**
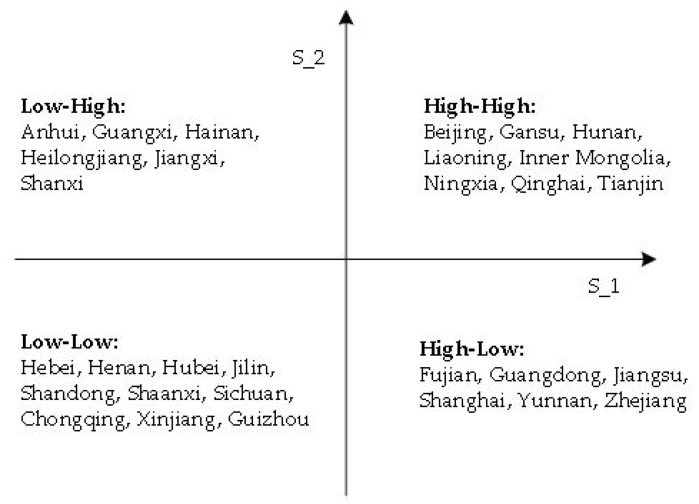
Province Distribution by Stage Efficiency from 2014 to 2017.

**Table 1 healthcare-08-00119-t001:** Input and Output Variables.

Stage	Variable		Unit
Wastewater Treatment Stage	Input	Sewage treatment plants	Number
	Output	Total wastewater discharge	10,000 tons
		Municipal sewage treatment capacity	10,000 CBM
		COD concentration	‰
		Heavy metal pollutant equivalent concentration (intermediate)	—
Carryover		Completed investment in wastewater treatment project	10,000 CNY
Health Stage	Input	Number of health technicians	10,000 people
		Local fiscal medical and health expenditure	100 million CNY
	Output	Average life expectancy	—
		Carcinogenic risk	—

**Table 2 healthcare-08-00119-t002:** Descriptive Statistics of Inputs and Outputs.

Variable	Average	Maximum	Minimum	Std. Dev.
Investment	34,779.3333	151,037.0000	697.2500	36,582.7239
Sewage treatment plants	66.5250	252.2500	10.2500	52.3666
Total wastewater discharge	238,315.7487	909,221.5500	25,263.5425	192,271.1729
Municipal sewage treatment capacity	145,230.0167	636,730.0000	12,268.2500	126,412.0661
COD concentration	0.2722	0.6708	0.0760	0.1348
Heavy metal pollutant equivalent concentration	10.3559	69.7938	0.1652	14.8889
Number of health technicians	27.4513	64.8700	3.7000	16.0865
Local fiscal medical and health expenditure	409.0873	1031.3250	79.8475	207.4266
Average life expectancy	75.7869	82.4363	70.1982	2.6148
Carcinogenic risk	56.8171	354.6568	1.1537	82.1253

**Table 3 healthcare-08-00119-t003:** Overall Efficiency by Provinces from 2014 to 2017.

Rank	DMU	Overall	2014	2015	2016	2017
1	Ningxia	1.0000	1.0000	1.0000	1.0000	1.0000
1	Qinghai	1.0000	1.0000	1.0000	1.0000	1.0000
3	Hainan	0.7792	1.0000	1.0000	0.6728	0.6566
4	Tianjin	0.7221	0.4940	0.7298	1.0000	0.7880
5	Beijing	0.6995	0.6128	0.6246	1.0000	0.6897
6	Hunan	0.6881	0.6261	0.7270	1.0000	1.0000
7	Gansu	0.6819	0.6800	0.7212	0.7447	0.8294
8	Inner Mongolia	0.5990	0.7142	0.6840	0.6338	0.6289
9	Liaoning	0.5811	0.5005	0.4266	1.0000	0.5220
10	Shanghai	0.5659	0.6057	0.6119	0.6299	0.6285
11	Guangxi	0.5068	0.4722	0.8034	0.4483	0.4770
12	Heilongjiang	0.4862	0.4012	0.4492	0.6608	0.4207
13	Fujian	0.4814	0.6681	0.6465	0.6293	0.3519
14	Jiangxi	0.4807	0.3937	0.9707	0.4845	0.3273
15	Anhui	0.4454	0.3629	0.4198	0.4054	0.7646
16	Yunnan	0.4446	0.5005	0.5450	0.5184	0.5070
17	Shanxi	0.4291	0.3487	0.3600	0.7558	0.2920
18	Xinjiang	0.4170	0.3664	0.3989	0.4379	0.4565
19	Zhejiang	0.4162	0.6225	0.6191	0.6424	0.4910
20	Chongqing	0.3815	0.3205	0.3154	0.5248	0.4011
21	Guangdong	0.3644	0.5815	0.5866	0.5653	0.6149
22	Jilin	0.3587	0.3512	0.3055	0.3664	0.3939
23	Hubei	0.3487	0.4796	0.3866	0.4494	0.3819
24	Jiangsu	0.3477	0.3888	0.4970	0.5555	0.6052
25	Guizhou	0.3374	0.2534	0.6627	0.2668	0.2346
26	Shaanxi	0.3267	0.3064	0.3522	0.3100	0.2972
27	Henan	0.3218	0.3008	0.3363	0.2895	0.6852
28	Shandong	0.2708	0.3005	0.3319	0.3552	0.3122
29	Hebei	0.2580	0.2551	0.2553	0.2797	0.2592
30	Sichuan	0.2575	0.2655	0.3647	0.2958	0.2397

**Table 4 healthcare-08-00119-t004:** Comparison of Two-stage Efficiency Scores from 2014–2017.

DMU	2014S_1	2014S_2	2015S_1	2015S_2	2016S_1	2016S_2	2017S_1	2017S_2
Anhui	0.4169	0.3090	0.4739	0.3657	0.5501	0.2607	0.5293	1.0000
Beijing	1.0000	0.2256	1.0000	0.2493	1.0000	1.0000	1.0000	0.3794
Fujian	1.0000	0.3363	1.0000	0.2930	1.0000	0.2587	0.4576	0.2462
Gansu	0.8924	0.4675	1.0000	0.4424	1.0000	0.4894	1.0000	0.6589
Guangdong	1.0000	0.1630	1.0000	0.1731	1.0000	0.1305	1.0000	0.2297
Guangxi	0.5987	0.3457	0.6067	1.0000	0.5336	0.3629	0.5708	0.3833
Guizhou	0.2783	0.2284	0.3253	1.0000	0.3300	0.2035	0.2595	0.2098
Hainan	1.0000	1.0000	1.0000	1.0000	0.3455	1.0000	0.3132	1.0000
Hebei	0.4109	0.0992	0.4059	0.1048	0.4275	0.1319	0.3973	0.1210
Henan	0.3998	0.2018	0.4776	0.1950	0.4686	0.1104	0.3705	1.0000
Heilongjiang	0.4976	0.3048	0.5962	0.3021	0.5402	0.7813	0.3743	0.4670
Hubei	0.7229	0.2362	0.5582	0.2151	0.6015	0.2974	0.6306	0.1333
Hunan	1.0000	0.2522	1.0000	0.4540	1.0000	1.0000	1.0000	1.0000
Jilin	0.4095	0.2930	0.3904	0.2207	0.4718	0.2609	0.4988	0.2891
Jiangsu	0.5649	0.2127	0.7894	0.2046	1.0000	0.1110	1.0000	0.2104
Jiangxi	0.4727	0.3147	0.9413	1.0000	0.6525	0.3166	0.4790	0.1756
Liaoning	0.6086	0.3925	0.6550	0.1983	1.0000	1.0000	0.7582	0.2858
Inner Mongolia	1.0000	0.4285	1.0000	0.3680	1.0000	0.2676	0.5394	0.7185
Ningxia	1.0000	1.0000	1.0000	1.0000	1.0000	1.0000	1.0000	1.0000
Qinghai	1.0000	1.0000	1.0000	1.0000	1.0000	1.0000	1.0000	1.0000
Shandong	0.4066	0.1944	0.4737	0.1900	0.5940	0.1164	0.5025	0.1219
Shanxi	0.4125	0.2849	0.5179	0.2020	0.5115	1.0000	0.3468	0.2372
Shaanxi	0.4268	0.1860	0.5514	0.1531	0.4201	0.1998	0.3648	0.2296
Shanghai	1.0000	0.2113	1.0000	0.2237	1.0000	0.2598	1.0000	0.2571
Sichuan	0.3844	0.1467	0.5509	0.1785	0.4416	0.1499	0.3469	0.1324
Tianjin	0.5261	0.4618	1.0000	0.4597	1.0000	1.0000	0.5761	1.0000
Xinjiang	0.4921	0.2407	0.5268	0.2710	0.4771	0.3988	0.3848	0.5282
Yunnan	0.7433	0.2576	0.8036	0.2863	0.7000	0.3369	0.6452	0.3687
Zhejiang	1.0000	0.2450	1.0000	0.2383	1.0000	0.2847	0.7205	0.2616
Chongqing	0.4047	0.2362	0.4165	0.2143	0.3804	0.6692	0.4042	0.3981

S_1 refers to wastewater treatment stage in DEA analysis; S_2 refers to health stage in DEA analysis.

**Table 5 healthcare-08-00119-t005:** Sewage Treatment Plants’ Efficiency of Each Province from 2014 to 2017.

DMU	2014	2015	2016	2017	DMU	2014	2015	2016	2017
Anhui	0.5628	0.6014	0.6694	0.6872	Jiangxi	0.5653	0.9913	0.8502	0.6205
Beijing	1.0000	1.0000	1.0000	1.0000	Liaoning	0.7596	0.7956	1.0000	0.8886
Fujian	1.0000	1.0000	1.0000	0.7105	Inner Mongolia	1.0000	1.0000	1.0000	0.6029
Gansu	0.8586	1.0000	1.0000	1.0000	Ningxia	1.0000	1.0000	1.0000	1.0000
Guangdong	1.0000	1.0000	1.0000	1.0000	Qinghai	1.0000	1.0000	1.0000	1.0000
Guangxi	0.9211	0.8265	0.6874	0.7581	Shandong	0.4602	0.5384	0.4618	0.4015
Guizhou	0.3814	0.4183	0.4008	0.2806	Shanxi	0.5484	0.6583	0.6146	0.4570
Hainan	1.0000	1.0000	0.4100	0.4163	Shaanxi	0.6087	0.8278	0.5847	0.5386
Hebei	0.5463	0.5268	0.4774	0.4541	Shanghai	1.0000	1.0000	1.0000	1.0000
Henan	0.6306	0.7306	0.5659	0.4441	Sichuan	0.5128	0.6876	0.4716	0.3640
Heilongjiang	0.6638	0.7223	0.4984	0.3511	Tianjin	0.5720	1.0000	1.0000	0.7542
Hubei	0.7274	0.6477	0.6297	0.6698	Xinjiang	0.4714	0.5270	0.4761	0.4152
Hunan	1.0000	1.0000	1.0000	1.0000	Yunnan	0.8410	0.9539	0.8767	0.6917
Jilin	0.4999	0.4613	0.4424	0.5850	Zhejiang	1.0000	1.0000	1.0000	0.8071
Jiangsu	0.4809	0.6838	1.0000	1.0000	Chongqing	0.5088	0.5022	0.4813	0.5381

**Table 6 healthcare-08-00119-t006:** Total Wastewater Discharge’ Efficiency of Each Province from 2014 to 2017.

DMU	2014	2015	2016	2017	DMU	2014	2015	2016	2017
Beijing	1.0000	1.0000	1.0000	1.0000	Hainan	1.0000	1.0000	0.6088	0.5048
Gansu	1.0000	1.0000	1.0000	1.0000	Xinjiang	0.8954	0.7470	0.8145	0.6342
Guangdong	1.0000	1.0000	1.0000	1.0000	Yunnan	0.6456	0.9542	0.8504	0.6157
Hunan	1.0000	1.0000	1.0000	1.0000	Jilin	0.6642	0.6635	0.8353	0.7531
Ningxia	1.0000	1.0000	1.0000	1.0000	Guangxi	0.5940	0.7070	0.7588	0.7427
Qinghai	1.0000	1.0000	1.0000	1.0000	Anhui	0.5530	0.6569	0.7742	0.7511
Shanghai	1.0000	1.0000	1.0000	1.0000	Shanxi	0.6257	0.7676	0.7299	0.5125
Inner Mongolia	1.0000	1.0000	1.0000	0.8934	Shandong	0.5837	0.6572	0.7338	0.6225
Fujian	1.0000	1.0000	1.0000	0.8082	Jiangxi	0.3800	0.9412	0.6872	0.4948
Tianjin	0.9004	1.0000	1.0000	0.7884	Shaanxi	0.5889	0.7590	0.6177	0.5168
Liaoning	0.8744	0.9935	1.0000	0.7823	Hebei	0.5927	0.6104	0.6099	0.6282
Zhejiang	1.0000	1.0000	1.0000	0.6339	Sichuan	0.5218	0.7266	0.6250	0.5342
Jiangsu	0.6556	0.8949	1.0000	1.0000	Chongqing	0.6350	0.6824	0.5389	0.5452
Heilongjiang	0.7443	1.0000	1.0000	0.7075	Henan	0.4487	0.5329	0.5254	0.4463
Hubei	0.7184	0.7522	0.8531	0.8781	Guizhou	0.4134	0.5202	0.5381	0.4614

**Table 7 healthcare-08-00119-t007:** COD Concentration Efficiency of Each Province from 2014 to 2017.

DMU	2014	2015	2016	2017	DMU	2014	2015	2016	2017
Beijing	1.0000	1.0000	1.0000	1.0000	Tianjin	0.2015	1.0000	1.0000	0.3223
Gansu	1.0000	1.0000	1.0000	1.0000	Liaoning	0.3153	0.2683	1.0000	0.7960
Guangdong	1.0000	1.0000	1.0000	1.0000	Hainan	1.0000	1.0000	0.0514	0.0589
Hunan	1.0000	1.0000	1.0000	1.0000	Henan	0.3001	0.3544	0.6712	0.5966
Ningxia	1.0000	1.0000	1.0000	1.0000	Sichuan	0.3083	0.4328	0.5171	0.4110
Qinghai	1.0000	1.0000	1.0000	1.0000	Guangxi	0.4695	0.4724	0.2896	0.3706
Shanghai	1.0000	1.0000	1.0000	1.0000	Anhui	0.3235	0.3445	0.4537	0.4187
Zhejiang	1.0000	1.0000	1.0000	1.0000	Xinjiang	0.2224	0.5815	0.2950	0.2729
Jiangsu	0.9882	1.0000	1.0000	1.0000	Hebei	0.2284	0.1980	0.4570	0.2761
Inner Mongolia	1.0000	1.0000	1.0000	0.5132	Shanxi	0.1538	0.2468	0.3714	0.2040
Fujian	1.0000	1.0000	1.0000	0.4024	Chongqing	0.1737	0.1559	0.3182	0.3197
Jiangxi	1.0000	1.0000	0.6437	0.6717	Jilin	0.1574	0.1188	0.2918	0.3171
Yunnan	1.0000	0.6255	0.5327	0.9736	Heilongjiang	0.1700	0.1112	0.2263	0.1722
Shandong	0.4330	0.4760	1.0000	0.9622	Shaanxi	0.1937	0.1221	0.1380	0.1066
Hubei	1.0000	0.4921	0.5347	0.5454	Guizhou	0.1442	0.1152	0.1547	0.1403

**Table 8 healthcare-08-00119-t008:** Number of Health Technicians’ Efficiency of Each Province from 2014 to 2017.

DMU	2014	2015	2016	2017	DMU	2014	2015	2016	2017
Hainan	1.0000	1.0000	1.0000	1.0000	Jiangsu	0.7048	0.7366	0.1228	0.6248
Ningxia	1.0000	1.0000	1.0000	1.0000	Liaoning	0.6105	0.1801	1.0000	0.2188
Qinghai	1.0000	1.0000	1.0000	1.0000	Hubei	0.7001	0.7722	0.3714	0.1299
Hunan	0.6707	0.6566	1.0000	1.0000	Beijing	0.3041	0.2387	1.0000	0.4180
Tianjin	0.6125	0.5881	1.0000	1.0000	Guizhou	0.2356	1.0000	0.2545	0.2444
Gansu	0.8962	0.9333	0.4810	0.6639	Shanxi	0.2283	0.1997	1.0000	0.2495
Guangxi	0.7723	1.0000	0.3741	0.8207	Chongqing	0.2930	0.3010	0.6643	0.4077
Anhui	0.9169	0.6647	0.3336	1.0000	Heilongjiang	0.2365	0.2478	0.7185	0.4254
Jiangxi	0.7321	1.0000	0.7362	0.2042	Shandong	0.5785	0.6212	0.1031	0.1001
Yunnan	0.4186	0.9861	0.8880	0.3739	Sichuan	0.2507	0.8140	0.1501	0.1523
Zhejiang	0.6708	0.6606	0.6407	0.6008	Xinjiang	0.2521	0.2485	0.3394	0.4666
Henan	0.7007	0.5977	0.1168	1.0000	Shanghai	0.2727	0.2919	0.3621	0.3575
Inner Mongolia	0.6201	0.7951	0.2425	0.7468	Jilin	0.2698	0.2692	0.2766	0.3520
Fujian	0.8530	0.7627	0.3297	0.3135	Shaanxi	0.1588	0.1581	0.2336	0.2395
Guangdong	0.5961	0.8107	0.1775	0.6058	Hebei	0.1152	0.1448	0.1580	0.1382

**Table 9 healthcare-08-00119-t009:** Local Fiscal Medical and Health Expenditure Efficiency of Each Province from 2014 to 2017.

DMU	2014	2015	2016	2017	DMU	2014	2015	2016	2017
Anhui	1.0000	0.6330	0.3453	1.0000	Jiangxi	0.7481	1.0000	0.7528	0.2652
Beijing	0.3516	0.2649	1.0000	0.4331	Liaoning	1.0000	0.3057	1.0000	0.3909
Fujian	1.0000	0.8257	0.3525	0.3393	Inner Mongolia	0.9177	1.0000	0.3558	0.6901
Gansu	1.0000	1.0000	0.4978	0.6539	Ningxia	1.0000	1.0000	1.0000	1.0000
Guangdong	0.7814	1.0000	0.2045	0.5836	Qinghai	1.0000	1.0000	1.0000	1.0000
Guangxi	1.0000	1.0000	0.3832	1.0000	Shandong	1.0000	1.0000	0.1586	0.1669
Guizhou	0.2212	1.0000	0.2704	0.2731	Shanxi	0.3415	0.2626	1.0000	0.3904
Hainan	1.0000	1.0000	1.0000	1.0000	Shaanxi	0.2170	0.2087	0.3302	0.3448
Hebei	0.1496	0.1801	0.2178	0.2068	Shanghai	0.2788	0.2950	0.3292	0.3384
Henan	1.0000	0.7914	0.1561	1.0000	Sichuan	0.3181	1.0000	0.1750	0.1836
Heilongjiang	0.3731	0.3565	0.8442	0.5918	Tianjin	0.5638	0.4998	1.0000	1.0000
Hubei	1.0000	1.0000	0.3797	0.2104	Xinjiang	0.3322	0.3215	0.4581	0.5897
Hunan	1.0000	1.0000	1.0000	1.0000	Yunnan	0.4538	1.0000	1.0000	0.3635
Jilin	0.3348	0.3166	0.3157	0.4565	Zhejiang	1.0000	1.0000	1.0000	1.0000
Jiangsu	1.0000	1.0000	0.1711	0.9140	Chongqing	0.3110	0.2912	0.6741	0.4235

**Table 10 healthcare-08-00119-t010:** Average Life Expectancy Efficiency of Each Province from 2014 to 2017.

DMU	2014	2015	2016	2017	DMU	2014	2015	2016	2017
Anhui	0.1921	0.3924	1.0000	1.0000	Jiangxi	0.2701	1.0000	0.2700	1.0000
Beijing	0.8226	1.0000	1.0000	1.0000	Liaoning	0.3222	1.0000	1.0000	1.0000
Fujian	0.2326	0.2640	1.0000	1.0000	Inner Mongolia	0.3863	0.3183	1.0000	1.0000
Gansu	0.3367	0.3272	1.0000	1.0000	Ningxia	1.0000	1.0000	1.0000	1.0000
Guangdong	0.1458	0.1066	0.5191	0.2394	Qinghai	1.0000	1.0000	1.0000	1.0000
Guangxi	0.2423	1.0000	1.0000	0.2667	Shandong	0.1405	0.1346	1.0000	1.0000
Guizhou	1.0000	1.0000	1.0000	1.0000	Shanxi	1.0000	0.9895	1.0000	1.0000
Hainan	1.0000	1.0000	1.0000	1.0000	Shaanxi	1.0000	1.0000	1.0000	1.0000
Hebei	1.0000	0.7839	1.0000	1.0000	Shanghai	1.0000	1.0000	1.0000	1.0000
Henan	0.1398	0.1633	1.0000	1.0000	Sichuan	0.3579	0.1100	1.0000	1.0000
Heilongjiang	1.0000	1.0000	1.0000	1.0000	Tianjin	1.0000	1.0000	1.0000	1.0000
Hubei	0.1709	0.1486	1.0000	1.0000	Xinjiang	1.0000	1.0000	1.0000	1.0000
Hunan	0.1778	0.3775	1.0000	1.0000	Yunnan	0.4190	0.1733	0.2172	1.0000
Jilin	1.0000	1.0000	1.0000	1.0000	Zhejiang	0.1931	0.1833	0.2200	0.2219
Jiangsu	0.1530	0.1338	1.0000	0.1767	Chongqing	1.0000	1.0000	1.0000	1.0000

**Table 11 healthcare-08-00119-t011:** Carcinogenic Risk Efficiency of Each Province from 2014 to 2017.

DMU	2014	2015	2016	2017	DMU	2014	2015	2016	2017
Anhui	1.0000	1.0000	0.3958	1.0000	Jiangxi	1.0000	1.0000	1.0000	0.3264
Beijing	0.3093	0.9799	1.0000	0.7569	Liaoning	1.0000	0.5499	1.0000	0.8666
Fujian	0.7877	0.3660	0.3629	0.3487	Inner Mongolia	1.0000	0.2633	0.7644	1.0000
Gansu	0.9143	0.6865	1.0000	1.0000	Ningxia	1.0000	1.0000	1.0000	1.0000
Guangdong	0.4110	0.9185	1.0000	1.0000	Qinghai	1.0000	1.0000	1.0000	1.0000
Guangxi	1.0000	1.0000	0.9134	1.0000	Shandong	1.0000	0.8995	0.7511	0.8108
Guizhou	1.0000	1.0000	0.4215	0.5330	Shanxi	1.0000	0.7226	1.0000	0.3023
Hainan	1.0000	1.0000	1.0000	1.0000	Shaanxi	0.9800	0.6032	0.1785	0.4552
Hebei	0.3306	0.1751	0.1507	0.1492	Shanghai	0.3903	0.3766	0.3392	0.2928
Henan	0.7261	1.0000	0.5265	1.0000	Sichuan	0.9168	0.9280	0.8317	0.4633
Heilongjiang	1.0000	1.0000	1.0000	0.8218	Tianjin	0.4525	0.6332	1.0000	1.0000
Hubei	0.6554	0.4917	0.4740	0.4459	Xinjiang	0.5720	0.8965	1.0000	1.0000
Hunan	1.0000	1.0000	1.0000	1.0000	Yunnan	1.0000	0.8333	1.0000	1.0000
Jilin	0.9362	0.3451	0.7298	0.2030	Zhejiang	0.3591	0.4854	0.7831	0.3870
Jiangsu	0.5214	0.9871	0.3526	0.3436	Chongqing	0.4435	0.2358	1.0000	0.9119

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
