# Peer review of "The Impacts of Water Pollution Emissions on Public Health in 30 Provinces of China"

_healthcare, 2020, doi:10.3390/healthcare8020119_

Round 1

Reviewer 1 Report

This is my second review of the article entitled Water Pollution Emissions impact on Public Health in Chinese 30 Provinces, written by Zhen Shi, Shijiong Qin, Chenjun Zhang, Yung-ho Chiu, Lina Zhang and submitted to Healthcare journal as a draft no 790567 (original one 729434) deals with an important issue of pollutants emission to environment.

Based on my experience with wastewater treatment, based on the parameters mentioned by the authors, I would not dare to make a diagnosis even for a single tank, which is a sewage receiver, not to mention the health of the entire population.

You can search for statistical relationships, but without understanding the meaning and the relationship between parameters, the results obtained are only numbers. They can be interpreted but the results are completely random.

The analyzed period of 2014-2017 is very short from an environmental point of view and in the context of persistence and non-degradability as well as bioaccumulation of pollutants should be analyzed chronic and not acute effects. It is completely not taken into account.

COD is general parameter. It is a sum of all substances that can react with potassium dichromate. The authors see no difference here between the harmful effects of e.g. glucose and benzo-a-pyrene, which is a catastrophic mistake. The authors in chapter 2.3 mention the decisive importance of other substances, but completely ignore them in the model.

The population's health is affected by many more factors than just the state of water / sewage quality. How are the authors able to distinguish the incidence related to 100% water quality, how do they exclude the influence of air quality, lifestyle etc.?

The Authors add 3 articles on the impact of sewage on the quality of the environment, indicating additional important indicators (e.g. total phosphorus) why these factors are not taken into account in the model (Wojtkowska and Bojanowski, 2018)? What about these deposits/precipitates (Deng et al 2019)? What does the impact of remediation of contaminated sites look like then? Is Shi et al, 2020 only article about environmental modeling, that could be used in discussion for this article? What about the others?

The harmful effects of heavy metals cannot be limited to hexavalent chromium, cadmium and arsenic (even assuming that arsenic is a heavy metal). We cannot overlook the effects of mercury, zinc, copper or lead.

In such articles, it makes no sense to refer to research on wastewater treatment conducted on a laboratory scale or discuss processes carried out in a laboratory. Instead, the results of industrial research and implementation works should appear that have a real impact on the quality of the environment.

As China is intensively developing its industry and putting more and more emphasis on the state of environmental quality, articles on Chinese implementation rams regarding wastewater treatment technology are published. Why do the authors not mention them?

The article clearly shows the work inserted by the authors, however, the results and calculations obtained by them are subject to great uncertainty and error. I do not see a key chapter describing these uncertainties and related discussions.

Author Response

Responses to Review 1

Thank you very much for your helpful feedback and insightful comments. We have taken all of the suggestions and comments into consideration during this revision. We truly appreciate the opportunity to revise our paper, and believe that our manuscript has significantly improved in response to the ideas and recommendations of the review team.

  1. Based on my experience with wastewater treatment, based on the parameters mentioned by the authors, I would not dare to make a diagnosis even for a single tank, which is a sewage receiver, not to mention the health of the entire population.You can search for statistical relationships, but without understanding the meaning and the relationship between parameters, the results obtained are only numbers. They can be interpreted but the results are completely random. The analyzed period of 2014-2017 is very short from an environmental point of view and in the context of persistence and non-degradability as well as bioaccumulation of pollutants should be analyzed chronic and not acute effects. It is completely not taken into account.

Our Response: Thank you for pointing this out. Wastewater treatment is a complex industrial process involving many chemical reactions, which is indeed a direction worth studying, but the output we focus on cannot reflect such complex wastewater treatment results and the physical and chemical residues of all reactants for the time being. In addition, pollutants have both chronic and acute effects on health in the context of their persistence and indegradability and bioaccumulation. However, due to the limited available data, this paper can only select relevant data from 2014 to 2017 to explore the impact of pollutants on health. We have apologized for these deficiencies and explained them in line 761-763 on page 24. Details are as follows.

The data of urban sewage treatment from 2014 to 2017 are selected in this paper. The research period is relatively short and the situation of sewage treatment in rural China isn’t taken into account. We will continue to follow up China's sewage treatment situation in the following period.

  1. COD is general parameter. It is a sum of all substances that can react with potassium dichromate. The authors see no difference here between the harmful effects of e.g. glucose and benzo-a-pyrene, which is a catastrophic mistake. The authors in chapter 2.3 mention the decisive importance of other substances, but completely ignore them in the model.

Our Response: Thank you for pointing this out. COD refers to the use of chemical oxidants (such as potassium dichromate) in water reducing substances (such as organic matter) and the oxidation decomposition of oxygen consumption. But in the wastewater, the substances that can react with potassium dichromate are mostly organic. According to the investigation in this paper, the organics in urban sewage are mainly harmful organics, such as benzo-a-pyrene, while the content of harmless organics such as glucose is relatively small, which can be neglected. Therefore, the effects of various organics are not compared in detail in this paper, and COD is used as the parameter to measure the organic pollution. The substances mentioned in chapter 2.3 have no significant significance for the study of this paper, so they are not applied to the model. We have further explained the concept of COD in line 436-444 on page 11-12 . Details are as follows.

⑤ COD concentration. It is defined as the concentration of oxygen required to oxidize organic pollutants in water with chemical oxidants. COD refers to the use of chemical oxidants (such as potassium dichromate) in water reducing substances (such as organic matter) and the oxidation decomposition of oxygen consumption, reflecting the extent of water pollution by reducing substances. The reducing substances can reduce the content of dissolved oxygen in the water, leading to the death of organisms in the water due to hypoxia and the deterioration of water quality. A higher COD denotes a higher content of reducing substances in the water and the more serious pollution. Since organic matter is the most common reducing substance in water, COD is an important parameter to measure organic pollution. 

  1. The population's health is affected by many more factors than just the state of water / sewage quality. How are the authors able to distinguish the incidence related to 100% water quality, how do they exclude the influence of air quality, lifestyle etc.?

Our Response: Thank you for pointing this out. Residents' health is affected by water / sewage quality, air quality, lifestyle and many other factors. This paper studied the impact of carcinogenic risk value on health, excluding other factors that may have an impact on residents' health. The carcinogenic risk values of total chromium emission, arsenic emission and cadmium emission were calculated by using specific formulas, so as to evaluate the impact of the carcinogenic risk values on the health of residents. We have further explained the formulas in line 471-482 on page 12. Details are as follows.

⑩ Carcinogenic risk. It calculates the carcinogenic risk value of total chromium emission, arsenic emission, and cadmium emission. The health risks of individual carcinogenic pollutants in multiple exposure pathways are as follows:

(7)

In the formula, Ri represents the health risk value of a single pollutant under various exposure pathways, CDI represents the exposure dose, Sf represents the carcinogenic slope factor of the pollutant, and the unit is mg·kg-1·d-1. The higher the Ri value is, the greater is the health risk of a carcinogen - that is, the higher the cancer probability of the pollutant. In concrete analysis, the maximum acceptable risk level of the International Council on Cancer (ICRP), 5*10-5, is usually taken as a reference value, which is interpreted as no more than five people per 10,000 are affected by the chemical with a new disease or cancer. The formula for calculating the total risk of various carcinogens is shown below.

(8)

Here, Rti represents the total health risk of all pollutants in all exposure pathways.

  1. The Authors add 3 articles on the impact of sewage on the quality of the environment, indicating additional important indicators (e.g. total phosphorus) why these factors are not taken into account in the model (Wojtkowska and Bojanowski, 2018)? What about these deposits/precipitates (Deng et al 2019)? What does the impact of remediation of contaminated sites look like then? Is Shi et al, 2020 only article about environmental modeling, that could be used in discussion for this article? What about the others?

Our Response: Thank you for pointing this out. In wastewater treatment stage, the input variable is sewage treatment plants, and the output variables are total wastewater discharge, municipal sewage treatment capacity, COD concentration and heavy metal pollutant equivalent concentration. The factors (e.g. total phosphorus) are mainly found in untreated wastewater,but we focus on treated wastewater, so these factors they are not included in our model (Wojtkowska and Bojanowski, 2018). In chapter 2.3, we mainly reference the exciting studies to illustrate the impact of water pollution caused by urban production and living. The treatment of these deposits/precipitates (Deng et al 2019) and the impact of remediation of contaminated sites are not the point of the chapter 2.3. Through literature review and research, we have found these deposits/precipitates are filtered out directly and then processed by professionals. Shi et al ( 2019 ) used a two-stage dynamic DEA model to study the impact of water pollution on the environment and national health, the study used the two-stage dynamic SBM DEA model. The environmental modeling used in Shi et al’s paper is useful to our model. The model is specified in chapter 3. We have supplemented the environmental modeling used in Shi et al’s paper in line 114-123 on page 3. Details are as follows.

Shi et al. (2019) [5] used a two-stage dynamic DEA model to study the impact of water pollution on the environment and national health. The authors divided variables into two stages. In the first stage (production), labor, energy, and water consumption were the input variables, and GDP was the desired output variable, while COD, CO2, and chromium emissions were undesired output variables. In the second stage (health), the local financial health expenditure and the number of health technicians were input variables. The health index and the population mortality rate were the desired output and the undesired output variables, respectively. Fixed asset investment was selected as the carryover indicator in both stages. The findings showed that urban sewage damaged the sustainable development of the environment and economy to a certain extent and also dragged down the degree of national health.

References

  1. Shi, Z., Wu,F., Huang, H., Sun, R., Zhang, L(2019). "Comparing Economics, Environmental Pollution    and Health Efficiency in China." Environmental Research and Public Health 16: 4827.

  1. The harmful effects of heavy metals cannot be limited to hexavalent chromium, cadmium and arsenic (even assuming that arsenic is a heavy metal). We cannot overlook the effects of mercury, zinc, copper or lead.In such articles, it makes no sense to refer to research on wastewater treatment conducted on a laboratory scale or discuss processes carried out in a laboratory. Instead, the results of industrial research and implementation works should appear that have a real impact on the quality of the environment.

Our Response: Thank you for pointing this out. According to China's environmental quality standard for surface water GB3838-2002, the heavy metal index includes 6 items: cadmium (Cd), lead (Pb), chromium (Cr), nickel (Ni), zinc (Zn) and copper (Cu). However, there are some essential elements to support life, such as Zn, Cu and so on. No matter the lack or surplus of these elements, they will affect human health. There are other heavy metal elements, such as cadmium, chromium, etc., which have obvious toxic effects. No matter how they get into the body, they will cause poisoning, leading to serious illness and even death. Based on the existing literature and the quality monitoring data of dalian's key drinking water sources from China Statistical Yearbook from 2007 to 2011, chemical carcinogens include hexavalent chromium, cadmium and arsenic. So in this paper, hexavalent chromium, cadmium and arsenic are used as the indexes affecting health. We have rewritten the concept of heavy metal pollutant equivalent concentration in the line 445-457 on page 12. Besides, we have deleted research on wastewater treatment conducted on a laboratory scale or discuss processes carried out in a laboratory, and added the results of industrial research and implementation works in the line 175-186 on page 4-5 (Liu et al, 2017). Details are as follows.

Membrane technology has become one of the important technologies for wastewater treatment in the printing and dyeing industry. Using literature metrology with National Knowledge Infrastructure (CNKI) and Web of Science (TM) (SCI), Liu et al. (2017) [14] studied the application status and prospect of membrane technology in wastewater treatment of printing and dyeing industry. The results showed that by 2015, the total capacity of the membrane technology in dyeing wastewater treatment in China was about 662,000 m(3).d(-1) and the number of applications was 128 (with capacity >= 500 m(3).d(-1)). Besides, "Ultrafiltration (UF) + 'reverse osmosis' (RO)" was the most widely applied process of membrane technologies in dyeing wastewater treatment, and the " membrane bioreactor (MBR) + RO" and "Continuous Membrane Filtration (CMF) + RO" were closely behind. Membrane technologyis a promising and important technology in the wastewater treatment of the printing and dyeing industry.

⑥ Heavy metal pollutant equivalent concentration. It is calculated on the basis of different heavy metal concentrations from China Environmental Statistics Yearbook. It refers to the degree of harm to the environment. The higher the equivalent concentration of pollution is, the greater is the degree of harm to the environment. According to China's environmental quality standard for surface water GB3838-2002, the heavy metal index includes 6 items: cadmium (Cd), lead (Pb), chromium (Cr), nickel (Ni), zinc (Zn) and copper (Cu). However, there are some essential elements to support life, such as Zn, Cu and so on. No matter the lack or surplus of these elements, they will affect human health. There are other heavy metal elements, such as cadmium, chromium, etc., which have obvious toxic effects. No matter how they get into the body, they will cause poisoning, leading to serious illness and even death. Based on the existing literature [49-51] and the quality monitoring data of dalian's key drinking water sources [52], chemical carcinogens include hexavalent chromium, cadmium and arsenic. So in this paper, hexavalent chromium, cadmium and arsenic are used as the indexes affecting health.

References

  1. Liu, L.,Cheng, XF,. Chen, X,. Zheng, L,. Shi, D,. Cao, M and Zhang, (2017). "Applications of membrane technology in treating wastewater from the dyeing industry in China: current status and prospect." Desalination and Water Treatment 77: 366-376.
  2. Lik H, Qian J Z, eta.(2008),” Environmental health risk assessment for the Groundwater quality in different villages near a land fill site,” 2008 Proceedings of Information Technology and Environmental System Sc-iences:1251-1256.
  3. Sipter E.(2008),” Human health risk assessment of toxic metals,” Hungary, Semmelweis Egyetem (Hungary):1-80.
  4. Jinbo (2013),” Environmental health risk assessment of drinking water in dalian ,” Liaoning normal university.
  5. Water resources bulletin of dalian municipal water bureau(2007-2011), Dalian municipal water bureau, http://www.swj.dl.gov.cn/ 

  1. As China is intensively developing its industry and putting more and more emphasis on the state of environmental quality, articles on Chinese implementation rams regarding wastewater treatment technology are published. Why do the authors not mention them?

The article clearly shows the work inserted by the authors, however, the results and calculations obtained by them are subject to great uncertainty and error. I do not see a key chapter describing these uncertainties and related discussions.

Our Response: Thank you for pointing this out. We have described municipal sewage treatment methods in chapter 2.2 and added an article on Chinese implementation rams regarding wastewater treatment technology(Liu et al, 2017)in the line 175-186 on page 4-5. We have apologized for these uncertainties and explained them in line 761-763 on page 24. Details are as follows.

Membrane technology has become one of the important technologies for wastewater treatment in the printing and dyeing industry. Using literature metrology with National Knowledge Infrastructure (CNKI) and Web of Science (TM) (SCI), Liu et al. (2017) [14] studied the application status and prospect of membrane technology in wastewater treatment of printing and dyeing industry. The results showed that by 2015, the total capacity of the membrane technology in dyeing wastewater treatment in China was about 662,000 m(3).d(-1) and the number of applications was 128 (with capacity >= 500 m(3).d(-1)). Besides, "Ultrafiltration (UF) + 'reverse osmosis' (RO)" was the most widely applied process of membrane technologies in dyeing wastewater treatment, and the " membrane bioreactor (MBR) + RO" and "Continuous Membrane Filtration (CMF) + RO" were closely behind. Membrane technologyis a promising and important technology in the wastewater treatment of the printing and dyeing industry.

The data of urban sewage treatment from 2014 to 2017 are selected in this paper. The research period is relatively short and the situation of sewage treatment in rural China isn’t taken into account. We will continue to follow up China's sewage treatment situation in the following period.

References

  1. Liu, L.,Cheng, XF,. Chen, X,. Zheng, L,. Shi, D,. Cao, Mand Zhang, ZX. (2017). "Applications of membrane technology in treating wastewater from the dyeing industry in China: current status and prospect." Desalination and Water Treatment 77: 366-376.

Reviewer 2 Report

The paper has been structurally improved. Now, it can be accepted for publication.

Author Response

Dear reviewer:

     Thank you very much for your approval. Your comments will be of great help to this article and my future writing.

      Yours sincerely

Zhen Shi

This manuscript is a resubmission of an earlier submission. The following is a list of the peer review reports and author responses from that submission.

Round 1

Reviewer 1 Report

The paper "Water Pollution Emissions impact on Public Health in Chinese 30 Provinces" is interesting for journal readers but the manuscript needs major revisions before further consideration.

The aim of the analysis should be evidenced in the abstract and introduction sections. The methodological section relative to Data Envelopment Analysis (DEA) is too technical and it requires further explanation foor a full comprehension of the analysis.

The literature section should be enriched in such a way that the role of innovation for sustainable development is identified (Aldieri et al., 2019a and 2019b).

The interpretation of the results need further discussion in terms of policy implications.

The conclusions section should be improve. In particular, the weaknesses of the analysis and the insights for future research should be evidenced.

References.

Aldieri L., Carlucci F., Vinci C. P. & Yigitcanlar T. (2019). Environmental innovation, knowledge spillovers and policy implications: A systematic review of the economic effects literature. Journal of Cleaner Production, http://doi.org/10.1016/j.jclepro.2019.118051.

Aldieri L., Kotsemir M. & Vinci C. P. (2019). The role of environmental innovation through the technological proximity in the implementation of the sustainable development. Business Strategy and the Environment, 29, 493-503.

Reviewer 2 Report

The Figure 3 could be larger, because it is not to much clear. No more suggestions or comments. It is well written. 

Reviewer 3 Report

This is a very nice paper on a very relevant topic. As such it deserves publication.

This said I would strongly suggest that the authors

  1. The early part of the paper up to page 6 could be greatly consolidated so that the reader does not have to wait so long for the beginning of the substance of the paper.
  2. There is a number of formatting issues that have to be addressed with the location of the tables as well as the spacing of the math parts etc.
  3. Have the paper edit by a native speaker. Although the context is very appealing there are just too many awkward passages.

Reviewer 4 Report

Article entitled Water Pollution Emissions impact on Public Health in Chinese 30 Provinces, written by Zhen Shi, Shijiong Qin, Chenjun Zhang, Yung-ho Chiu, Lina Zhang and submitted to Healthcare journal as a draft no 729434 deals with an important issue of pollutants emission to environment.

While reading I found some statements missing, confusing or unclear. Below I enclose a list of my comments.

As English is not my native language, I am not able to assess language correctness.

All abbreviations used should be explained. All typos should be corrected.

Liens 106-107 vs lines 138-139 repetition.

WWTP are not related to gaseous substances only. Primarily they point source of water pollutants. It should be clearly pointed out what are those pollutants.

Literature review is chaotic. Mostly not dealing with water pollution and impact on water environment. No literature about modeling was provided. No specific substances were described. No description of treatment technologies was given.

COD can not be a risk factor it is just collective content of substances that react with potassium dichromate. It cannot see a difference between sulfide, glucose or a toxic organic compound! The Authors provide even incorrect COD definition (line 431-432).

Why do the authors consider all heavy metals to be equally harmful? There is no difference between iron and mercury?

What is the source for model’s data line 355, 377, 378?

The Authors run very complex model without crucial input data!

Drawing conclusions about the health of the population or the quality of the environment only on the basis of the average content of heavy metals and COD in wastewater is a catastrophic mistake. It means a total misunderstanding of the topic of environmental pollution control and the parameters used to describe it.